# Endothelial cell responses in sepsis are attenuated by targeting truncated procalcitonin

Laura Brabenec [1,2], Katharina EM Hellenthal [1,3], Sebastian Kintrup[1],
Laura Cyran[4], Philipp Burkard [4], Astrid Nottebaum [5], Klaus Schughart [6,7],
Stefan Wagner[8], Roland Arnold [8], Vera Rauschenberger[9],
Stefanie Kampmeier[9,10], Patrick Meybohm[4], Nicolas Schlegel [11],
Dietmar Vestweber [5] & Nana-Maria Wagner [4,12] ✉

Sepsis is associated with hypotension, vascular leakage, vasoplegia and microvascular dysfunction. Therefore, the endothelium is a target for sepsis therapies. Since truncated procalcitonin exerts vascular activity, we here evaluated the efficacy of targeting procalcitonin for vascular integrity and sepsis outcomes. Sepsis up-regulated >2000 genes involved in pro-inflammatory responses while similar numbers of genes involving cell growth and maintenance were down-regulated. Transcriptomic changes in endothelial cells diminished by >50% by anti-procalcitonin antibodies and this was functionally associated with preserved vascular barrier integrity in lungs and intestines, reduced sepsis-induced vasoplegia, preserved endothelial nitric oxide bioavailability, improved organ integrity and reduced sepsis severity in mice. Mechanistically, procalcitonin neutralization was associated with reduced signaling of the interleukin-17 pathway. We here show sepsis induces substantial changes to the endothelial transcriptome and vascular integrity and neutralizing procalcitonin is a suitable means to preserve endothelial homeostasis at a transcriptomic and functional level that could translate into organ protection during sepsis.

Sepsis is common and associated with high mortality[1]. The presence of (mostly) bacterial specimen in the blood stream leads to a dysregulated immune response that affects vascular function and organ integrity. Despite numerous efforts in identifying immune signatures and patient phenotypes, the primary care bundle for patients with sepsis is still limited to timely delivery of antibiotics and treatment of hypotension[2]. Hypotension is a cardinal symptom of patients with sepsis that mirrors a severe, ubiquitous vascular barrier defect

[1]University Hospital Münster, Department of Anaesthesiology, Intensive Care and Pain Medicine, Münster, Germany. [2]Department of Physiology and Pharmacology, Karolinska Institutet, Stockholm, Sweden. [3]Division of Pulmonary and Critical Care Medicine, Massachusetts General Hospital, Harvard Medical School, Boston, MA, USA. [4]University Hospital Würzburg, Department of Anaesthesiology, Intensive Care, Emergency and Pain Medicine, Würzburg, Germany. [5]Max-Planck Institute of Molecular Biomedicine, Department of Vascular Cell Biology, Münster, Germany. [6]University of Münster, Institute of Virology, Münster, Germany. [7]University of Tennessee Health Science Center, Department of Microbiology, Immunology and Biochemistry, Memphis, TN, USA. [8]University of Birmingham, Institute of Cancer and Genomic Sciences, College of Medical and Dental Sciences, Birmingham, UK. [9]University Hospital Würzburg, Institute for Hygiene and Microbiology, Würzburg, Germany. [10]University Hospital Würzburg, Infection Control and Antimicrobial Stewardship Unit, Würzburg, Germany. [11]University Hospital Würzburg, Department of General, Visceral, Transplantation, Vascular and Paediatric Surgery (Department of Surgery I), Würzburg, Germany. [12]Department of Anesthesiology, University Medical Center of the Johannes Gutenberg-University Mainz, Mainz, Germany. ✉e-mail: nmwagner@uni-mainz.de

resulting in fluid extravasation into tissues and intravascular hypovolemia. Hypovolemia and hypotension critically affect organ perfusion and endanger for the often detected sepsis-induced multiple organ dysfunction syndrome (MODS). At the level of the microcirculation, interstitial edema compromises capillary perfusion and tissue oxygenation that further aggravates cellular injury and organ dysfunction. This importance of the vasculature in maintaining macro-hemodynamics, microcirculatory function and thus organ integrity makes the endothelium an attractive target for limiting sepsis-induced complications and death[3,4]. Loss of endothelial integrity and function are crucial players during the development of sepsis-associated organ dysfunction[5–7]. However, little is known about the characteristics of the endothelium in sepsis and the possible impact of therapeutic strategies on endothelial profiles during severe systemic inflammation.

Procalcitonin is a 116 amino acid peptide hormone with mere ubiquitous expression in inflamed tissue[8]. During systemic inflammation, procalcitonin blood concentrations rapidly increase and decrease upon sufficient focus control (i.e., reduction of bacteremia due to antibiotic delivery). This dynamic makes procalcitonin a sufficient biomarker for the clinical workup of sepsis patients[9]. Until a few years ago, the biological role of procalcitonin itself was unclear. Then, we discovered that procalcitonin exerts direct effects on the endothelium following activation by cleavage of two N-terminal amino acids[10]. These procalcitonin isoforms have also been found in humans and under inflammatory conditions found in equal parts[11,12]. Following binding to the calcitonin receptor-like receptor (CRLR) in complex with the receptor activity modifying protein 1 (RAMP1), procalcitonin resulted in phosphorylation of endothelial adherens junction proteins thus inducing loss of barrier function that perpetuated the above mentioned mechanisms of sepsis-induced organ dysfunction. In turn, inhibition of procalcitonin activation conserved vascular integrity and reduced interstitial edema in septic mice and resulted in augmented microcirculation, reduced hypovolemia and vasopressor requirements in patients with systemic inflammation. These results suggested that targeting procalcitonin could be a means to directly preserve vascular function during sepsis.

So far, most therapeutic strategies failed to improve sepsis outcome[13,14], including antagonizing IL-1[15], TNFalpha[16] or application of hydrocortisone[17]. Therapeutic strategies for the treatment of septic patients may be successful when a punctual therapeutic intervention induces secondary effects that inhibit the vicious circle of tissue ischemia, injury and further immune activation during systemic inflammation. We here focused on the endothelium as a therapeutic target in sepsis and performed an in-depth characterization of the endothelial transcriptome in septic mice. We then evaluated the overall effects of antagonizing hyperprocalcitonemia in septic mice on endothelial transcriptomic profiles by synthesis of antibodies specifically targeting activated murine procalcitonin. Finally, we evaluated antibody-induced effects on endothelial transcriptomic profiles for relevance for the occurrence of vasoplegia and capillary leakage, the hallmarks of vascular dysfunction in patients on intensive care units.

## Results

### Transcriptome of murine endothelial cells in sepsis

To evaluate the changes induced by sepsis at the transcriptomic level in murine endothelial cells, we subjected wild type mice to the translational model of polymicrobial sepsis due to cecal puncture and peritonitis. As the lung is the most susceptible organ to sepsis-induced tissue injury[18–20], we particularly focused on the evaluation of pulmonary endothelial cells (Fig. 1a). RNA sequencing of pulmonary endothelial cells 18 h after cecal puncture revealed that sepsis induced up- and downregulation of various genes in the endothelium (Fig. 1b). Principal component analysis of $n = 5$ mice/group revealed close clustering of septic mice in contrast to both sham-operated and untreated mice (Fig. 1c). The most regulated genes are highlighted in Fig. 1d, e. Further characterization of up- and down-regulated pathways showed that endothelial cells upregulate various pathways associated with inflammation, in particular cytokine production and interaction with immune cells (Fig. 1f). In contrast, pathways associated with cell growth, angiogenesis and assembly of cellular junctions were down-regulated (Fig. 1g). In addition we subjected mice to procalcitonin injection and analyzed *VEGF* gene expression in freshly isolated murine pulmonary endothelial cells 18 h after injection. According to gene expression analysis, we found procalcitonin induced a reduction in *VEGFa* and *VEGFc* expression (Fig. 1h).

### Human gene expression analysis

In response to septic patients plasma, human pulmonary microvascular endothelial cells show higher expression of *Calca*, the gene encoding procalcitonin (PCT). This effect was abolished by procalcitonin antibodies, indicating that procalcitonin itself can perpetuate *Calca* expression and that endothelial cells are a source of procalcitonin. In contrast, *Calcrl* and *Ramp1* remained unchanged in endothelial cells (Fig. 1i-k). In line with recent literature[21], we found VEGF mRNA level was downregulated in human pulmonary endothelial cells when exposed to septic patient plasma with hyperprocalcitonemia (Fig.1l). We then related our findings to results previously obtained in humans. After 8 h of LPS exposure, 416 unique genes were regulated, 243 up- and 173 down. 331 of these DEGs were also annotated in our datasets. Of these LPS-DEGs, 169 overlapped with DEGs from the contrast of CLP versus controls and 118 overlapped to the DEGs from the contrast CLP plus AB to controls. These results showed that our findings were very similar to the human LPS-induction study. More importantly, AB-treatment suppressed these responses. We then looked in detail at the top 50 up-regulated DEGs from the 8 h LPS study. 36 of the LPS-DEGs were also annotated in our dataset. As shown in Fig. 1m, almost all LPS-DEGs genes were also up-regulated in our study, and all of these DEGs (except one) showed reduced expression in CLP plus AB treated samples. Again, these findings showed that our results were very similar to findings in LPS-treated human cells, and demonstrated an effect of AB treatment.

### Neutralizing PCT with an antibody targeting truncated PCT

A hallmark of sepsis is capillary leakage due to endothelial barrier breakdown which results from downregulation and phosphorylation of adherens junctions such as vascular endothelial cadherin (VE-cadherin) at tyrosine 685. With the plasma from mice previously subjected to transcriptomic analysis, we first verified the opening-effect of adherens junctions due to VE-cadherin phosphorylation induction in murine endothelial cells (from 1.0 to $1.9 \pm 0.3$-fold, $P < 0.05$, $n = 5$/group, Fig. 2a, b). As we had previously identified procalcitonin to be a potent mediator of VE-cadherin disassembly, we next confirmed hyperprocalcitonemia in septic mice (increase from $1.2 \pm 0.2$ ng/mL in sham-operated to $9.5 \pm 0.4$ ng/mL in septic animals 18 h after sepsis induction, $P < 0.001$, $n = 5–18$ mice/group, Fig. 2c). Procalcitonin exists in two forms, a full-length, 116 amino acid variant isoform and a truncated variant isoform, where dipeptidyl-peptidase 4 (DPP4) cleaves two amino acid at procalcitonins N-terminus. Since only the truncated variant isoform exerts activity on the vasculature[10], we designed a polyclonal antibody targeting the truncated N-terminus of procalcitonin (Fig. 2d). Using recombinant synthesized procalcitonin variant isoforms, we then confirmed antibodies targeting procalcitonins N-terminus of the truncated variant isoform sufficiently inhibit procalcitonin-induced VE-cadherin phosphorylation at tyrosine 685 (Fig. 2e). In addition the antibody antagonized loss of endothelial barrier function following application of septic mouse plasma on endothelial monolayers in vitro (from $1.7 \pm 0.2$-fold to $1.0 \pm 0.03$-fold, $P < 0.01$, $n = 4–8$ mice/group, Fig. 2f).

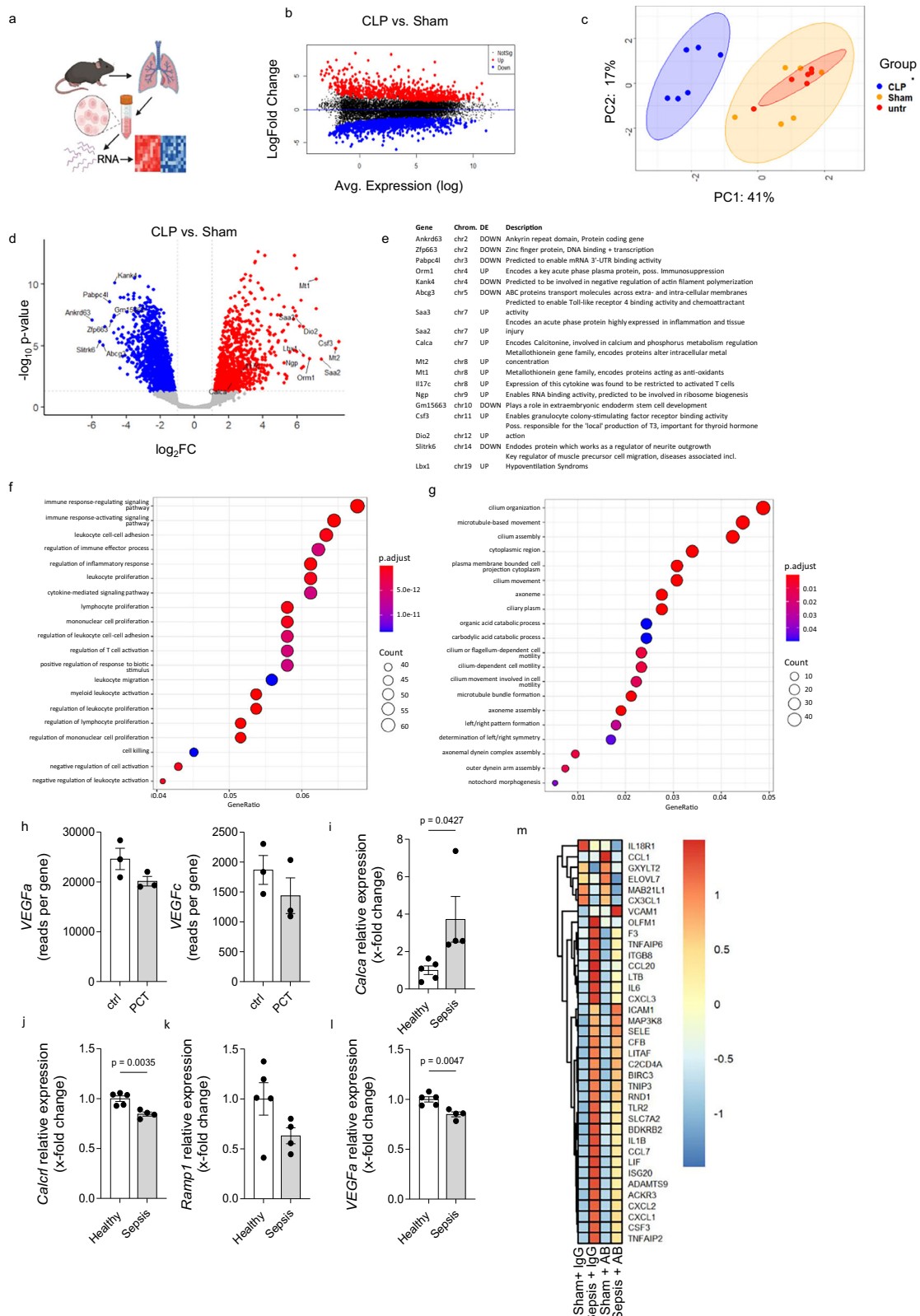

## Neutralizing PCT ameliorates endothelial activation

Having confirmed functional antagonization of procalcitonin effects by the neutralizing antibody in vitro, we evaluated the potency of neutralizing procalcitonin on modulating sepsis-induced changes to the endothelial transcriptome. Strikingly, antibody application was associated with a marked reduction of differentially expressed genes (DEGs) in septic animals (Fig. 3a). Antagonizing procalcitonin led to a reduction

of the number of differentially regulated genes in septic mice endothelium by approximately 50% in accordance with the findings shown in Fig. 3A (Fig. 3b–d). For example, we found sepsis to induce the expression of the interleukin-17 (IL-17) a and f genes (Supplementary Fig. 3). KEGG-pathway analysis revealed antibody-mediated downregulation of IL-17 and the IL-17 signaling pathway genes (Fig. 3e). This finding was confirmed at the protein expression level. While sepsis induced a 100-

**Fig. 1 | Transcriptome of murine endothelial cells in sepsis. a** Endothelial cells were isolated from murine lungs and subjected to bulk RNA sequencing. Created in BioRender. Brabenec, L. (https://BioRender.com/mpa96bi) **b** MDA plot showing DEGs in color that were more than 2-fold differentially regulated ($p < 0.05$) in mice after cecal ligation and puncture (CLP) vs. mice subjected to laparotomy only (sham). Limma uses an empirical Bayes shrinkage method to moderate the standard errors of the estimated log-fold changes, which includes t-tests foreach gene To control for multiple testing the Benjamini and Hochberg method was used. **c** Principal component analysis of normalized expression values for $n = 6$ mice/ group 18 h after sepsis induction. **d** Volcano plot representing DEGs in color and labelling of the 10 most up- and down-regulated genes in murine pulmonary endothelium. Limma uses an empirical Bayes shrinkage method to moderate the standard errors of the estimated log-fold changes, which includes t-tests foreach gene To control for multiple testing the Benjamini and Hochberg method was used. **e** List of the 10 most up- and down-regulated genes including the procalcitonin encoding gene *Calca*. **f** Up- and (**g**) down-regulated pathways in murine endothelium in response to polymicrobial sepsis. Enrichment analysis, statistical significance was assessed using adjusted *p* values (FDR correction) to account for multiple testing. **h** *VEGFa* and *VEGFc* is downregulated in isolated murine pulmonary endothelial cells 18 h after injection of human procalcitonin in mice, $n = 3$.

**i** Human pulmonary microvascular endothelial cells show increased *Calca* expression after treatment with sepsis patients serum. This could be abolished by the use of procalcitonin antibody, $n = 4$ (Healthy), $n = 5$ (Sepsis), $p = 0.0427$. **j, k** *Calcrl* and *Ramp1* show no difference in gene expression, $n = 4$ (Healthy), $n = 5$ (Sepsis), $p = 0.0035$. **l** *VEGFa* was downregulated in human pulmonary microvascular endothelial cells when treated with serum of septic patients, $n = 4$ (Healthy), $n = 5$ (Sepsis), $p = 0.0047$. **m** Expression of genes from LPS-treated human cells in our CLP study. We then took the top 50 up-regulated DEGs from the 8 h LPS study and found that 36 of these were also annotated in our dataset. Mean expression values for these genes from our study were calculated per group and then presented as heatmap with values scaled by row. Cluster 1: All genes in this cluster were up-regulated upon CLP as for LPS 8 h, and almost all DEGs (except *ICAM1*) were reduced in expression after AB treatment. Cluster 2: Genes in this cluster were down-regulated in our dataset upon CPL, but not up-regulated as in the LPS study. Of note, these DEGs were stronger down-regulated after AB treatment. One LPS-DEG gene, *VCAM1*, was also up-regulated after CLP, but expressed higher in AB-treated samples compared to non-AB-treated CLP controls. Unpaired t test, data presented as mean ± SEM. *p* values as indicated. Source data are provided as a Source Data file.

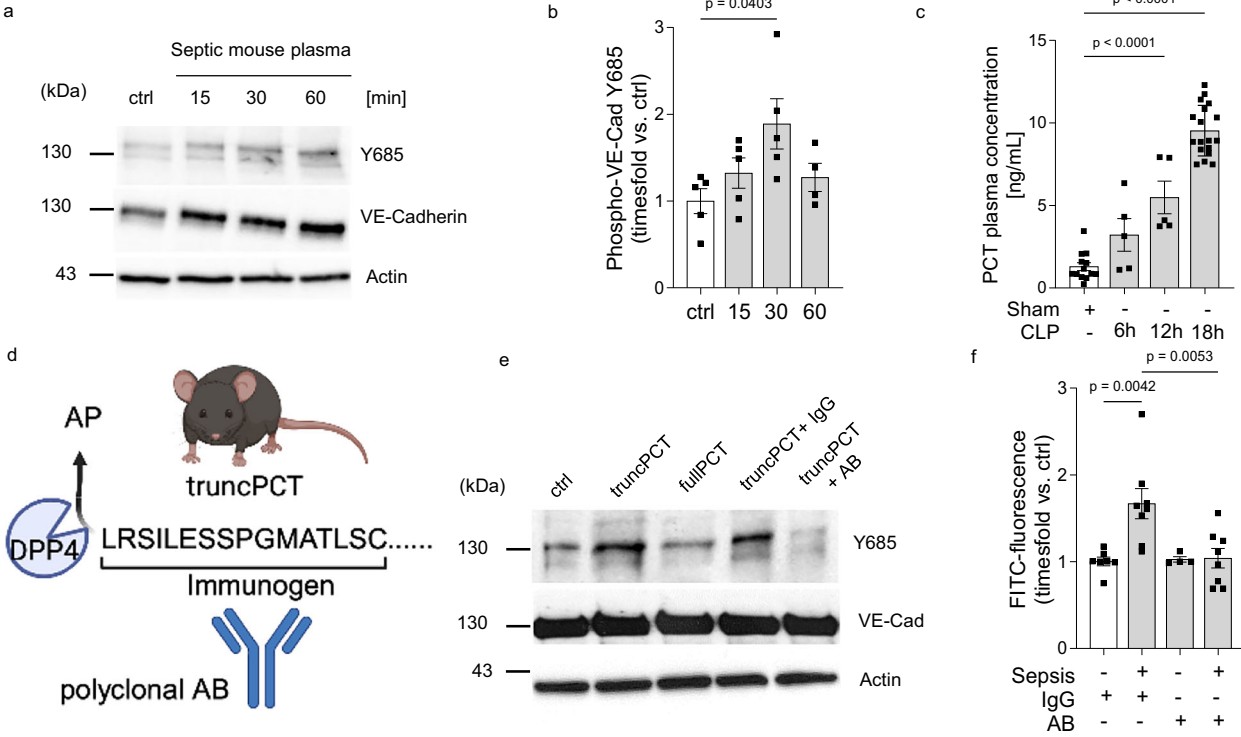

**Fig. 2 | Neutralizing procalcitonin with a polyclonal antibody targeting truncated procalcitonin. a, b** Representative immunoblot and quantitative summary showing phosphorylated VE-cadherin at tyrosine residue 685 in murine endothelial cell lysates after stimulation with septic mice plasma for 15, 30 and 60 min, $n = 4$ (60), $n = 5$ (ctrl,15,30) independent experiments/group, timesfold vs. control. Samples derive from the same experiment and blots were processed in parallel. Uncropped blots in Source Data. **c** Procalcitonin plasma concentrations during the course of sepsis in mice, $n = 5$ (CLP12h, 6 h), $n = 12$ (sham), $n = -18$ (CLP18h). **d** Schematic depicting the strategy of antibody generation targeting murine truncated procalcitonin. Created in BioRender. Brabenec, L. (https://BioRender.com/7xr1vmf) **e** Representative

immunoblot showing phosphorylated VE-cadherin at tyrosine residue 685 in endo-thelial cell lysates 30 min after exposure to recombinant procalcitonin, $n = 7$, timesfold vs. control. Samples derive from the same experiment and blots were processed in parallel. Uncropped blots in Source Data. **f** Fluorescence-labeled macromolecule permeability of murine pulmonary endothelial cells exposed to septic mice's plasma after pre-treatment with 1 µg of the antibody directed against truncated procalcitonin (PCT AB) and respective IgG control for two h, $n = 48$ (ctrl AB), $n = 7$ (ctrl IgG), $n = 8$ (Sepsis IgG, Sepsis AB). Data presented as mean ± SEM. One-way ANOVA/Bonferroni. *p* values as indicatedSource data are provided as a Source Data file.

fold increase in IL-17 (from 0.9 ± 0.4 to 100.2 ± 46.3 pg/mL, $P < 0.001$, $n = 6$ mice/group), neutralization of truncated procalcitonin (to 2.7 ± 1.0 pg/mL, $P < 0.001$), application of olcegepant, a procalcitonin receptor agonist during sepsis (to 3.4 ± 2.4 pg/mL, $P < 0.001$) as well as application of an inhibitor of dipeptidyl-peptidase 4 (DPP4) that inhibits

the conversion of full-length to truncated procalcitonin (to 0.7 ± 6.3 pg/ mL, $P < 0.001$) all nearly completely abolished the presence of IL-17 in septic mice plasma (Fig. 3f). Downregulation of various cytokines and inflammatory mediators in line with the genomic data was also observed (Fig. 3g).

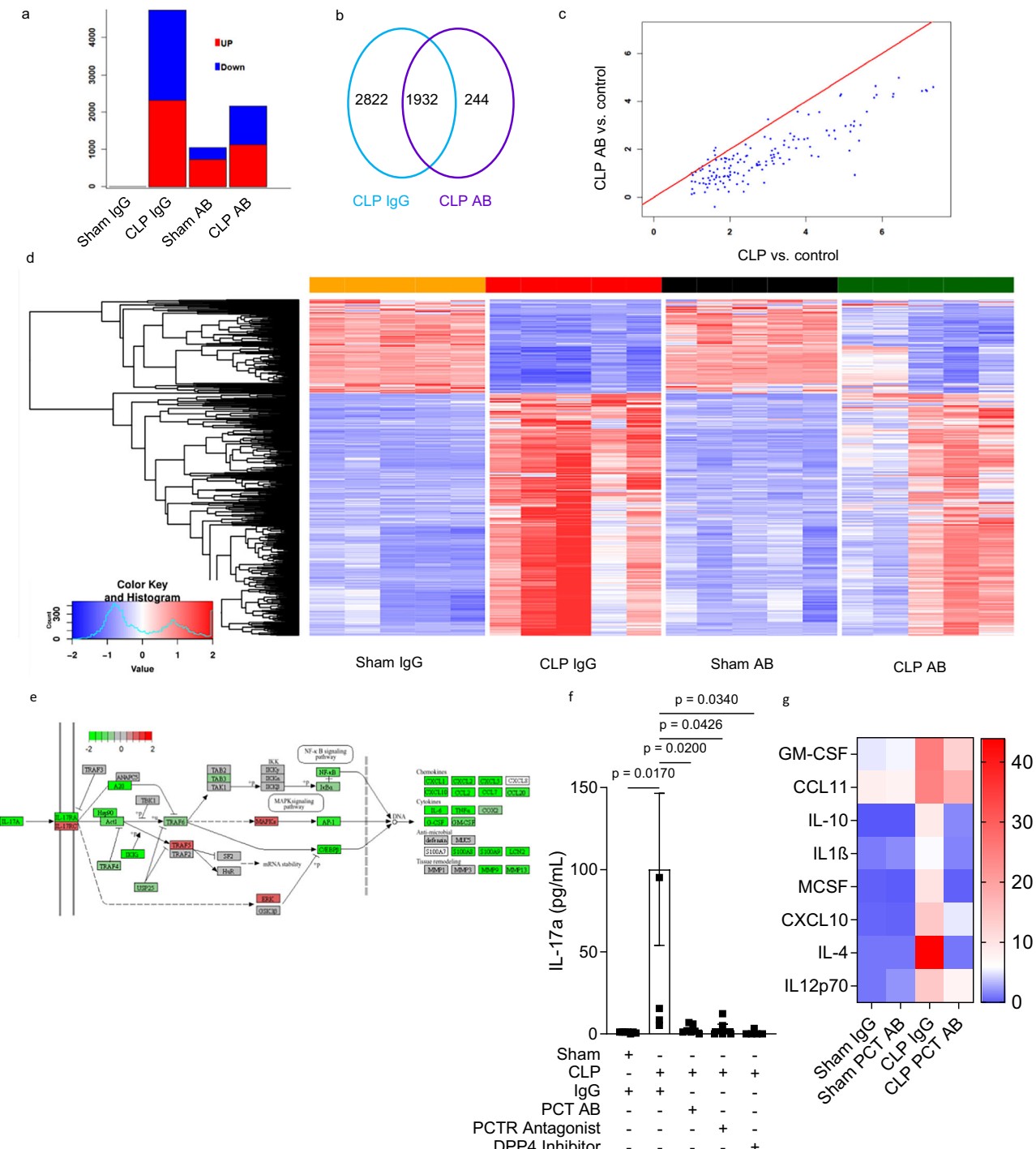

**Fig. 3 | Neutralizing procalcitonin ameliorates sepsis-induced endothelial activation in vivo. a** Summary of the number of up- (red) and down-(blue) differentially regulated genes (DEGs) in murine endothelial cells. Results are derived from relating sham AB to sham IgG [left column], CLP IgG to sham IgG, CLP AB to CLP IgG and CLP AB to sham AB, respectively. **b** VENN diagram deciphering the number of DEGs from the same contrasts. **c** Scatter plot of genes showing mean log$_2$ fold differences of CLP controls and CLP AB treated mice to the respective sham-treated controls. For this comparison, the up-regulated genes of the cytokine-mediated signaling pathway were used that were commonly up-regulated DEGs in CLP IgG and CLP AB treated versus the respective sham controls. **d** Heatmap of expression levels of DEGs from all contrasts. Values were scaled by row. red: up-regulated DEGs, blue: down-regulated DEGs. Each column represents one mouse. **e** Expression differences in *Il17* pathway. Colors represent differences

in normalized expression levels from the strongest CLP AB responder (sample CLP_PCT_AB_11_02) versus a strong CLP IgG responder (sample CLP_IgG_ctrl_16_07) projected on the mouse IL-17 signaling pathway (KEGG pathway mmu04657). **f** Inhibition of procalcitonin activation by DPP4 inhibitor sitagliptin and blocking procalcitonins receptor by the PCTR antagonist olcegepant, reduced IL-17 plasma levels in septic mice. $n = 7$ (Sham IgG, CLP PCTAB), $n = 6$ (CLP IgG), $n = 5$ (PCTS Antagonist, DPP4 Inhibitor), timesfold vs. control. **g** Heatmap showing cytokine and chemokine expression profiles in murine blood 18 h after sepsis induction by cecal ligation and puncture (CLP)/control (sham) surgery following injection of the antibody and respective control IgG 6 h after surgery, data shown as ng/mL, $n = 7$ mice/group. Data is presented as mean ± SEM. One- and Two-way ANOVA and Bonferroni-correction. $p$ values as indicated. Source data are provided as a Source Data file.

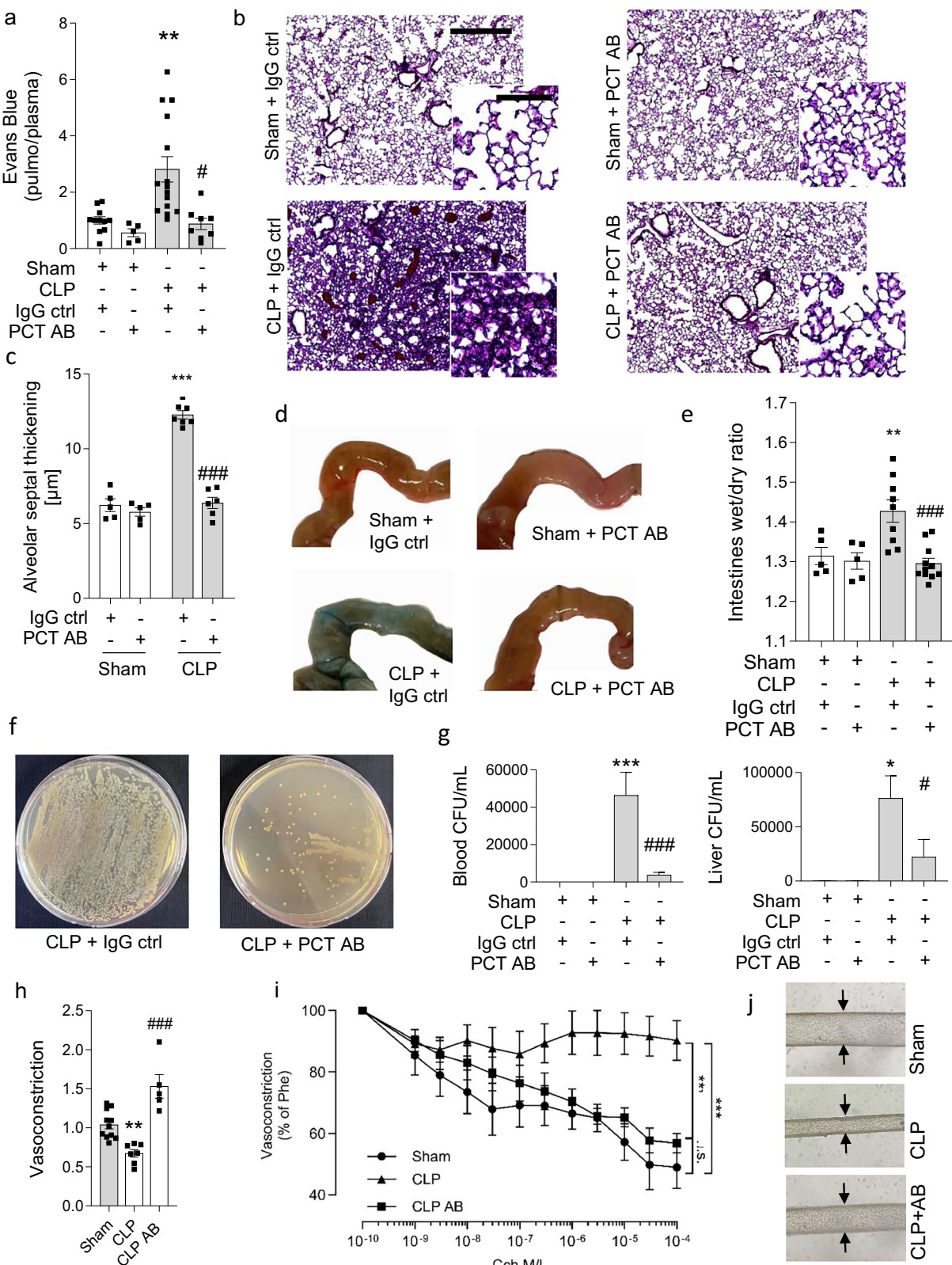

## Neutralizing PCT protects of endo- and epithelial barriers

We next aimed to dissect whether the reduced genomic activation and concentration of humoral mediators translates into the preservation of the vascular barrier integrity during sepsis. While septic mice exhibited fulminant pulmonary edema, extravasation of Evans blue-bound albumin (from $0.9 \pm 0.2$ pulmo/plasma ratio to $2.8 \pm 0.4$, $P < 0.05$, n = 5-14 mice/group) as well as fluid accumulation in the pulmonary

interstitium (from $6.4 \pm 0.4\,\mu m$ to $12.3 \pm 0.3\,\mu m$ of alveolar thickening, indicative of interstitial pulmonary edema, $P < 0.001$, n = 6 mice/group) was diminished with antibody application (to $0.9 \pm 0.2$ pulmo/plasma ratio and $6.3 \pm 0.7\,\mu m$, $P < 0.001$, respectively, Fig. 4a–c). Importantly, barrier protection was also evident in other organs such as the intestines, where application of the procalcitonin targeting antibody also reduced wet/dry ratios of murine guts (from

**Fig. 4 | Neutralizing procalcitonin exerts vascular protection. a** Pulmonary edema formation, as assessed by quantification of albumin-bound Evans blue dye, 18 h after sepsis induction by cecal ligation and puncture (CLP)/control (Sham) surgery following injection of the antibody and respective control IgG directly and 6 h after surgery $n = 5$(sham+AB), $n = 8$ (CLP + AB), $n = 11$ (Sham+ABctrl), $n = 15$ (CLP +ABctrl) mice/group. **b** Pulmonary edema visualized by hematoxylin and eosin-stained micrographs of murine lungs as indicated and **c** quantitative summary of alveolar septal thickening 18 h after CLP/sham surgery following antibody treatment. $n = 5$ (sham+ctrl, Sham+AB), $n = 6$ (CLP + AB) $n = 7$ (CLP+ctrl) mice/group. Scale bar indicates 1000 μm, scale bar for the inserts 100 μm. **d** Representative photographs showing Evans blue content of murine intestines presenting albumin uptake that indicates tissue edema formation. Scale bar indicates 5 mm. **e** Tissue edema of murine intestines after sepsis induction and treatment with the pro-calcitonin antibody or respective control IgG indicated as tissue wet/dry ratio 18 h after sepsis induction and antibody treatment, $n = 5$ (Sham+ctrl, Sham+AB), $n = 9$ (CLP+ctrl), $n = 11$(CLP + AB) mice/group. **f** Representative photographs showing agar plates of murine blood after incubation at 37 °C for 24 h. Scalebar indicates 10 mm. **g** Colony forming unit count in murine blood (left) and liver (right) culture withdrawn 18 h after sepsis induction by cecal ligation and puncture (CLP)/control (Sham) following injection of the antibody and respective control IgG directly and 6 h after surgery, $n = 4$(CLP+ctrl), $n = 5$ (sham+ctrl, sham+AB), $n = 9$ (CLP+ctrl), $n = 11$ (CLP + AB), $n = 16$ (CLP + AB) mice/group. **h** Catecholamine-responsiveness to 10 nM phenylephrine hydrochloride of third-order mesenteric resistance arteries 18 h after sepsis induction. $n = 5$ (CLP + AB), $n = 7$ (CLP), $n = 11$ (Sham) mice/group. **i** Vasodilatation to increasing concentrations of carbamoylcholine ($10^{-9}$ to $10^{-4}$ M) and **j** representative pictures showing vasodilation of mesenteric arteries.Scale bar indicates 50 μm. $n = 4$ (CLP), $n = 5$ (Sham), $n = 6$ (CLP + AB) mice/group. Data is presented as mean ± SEM. One- and Two-way ANOVA and Bonferroni-correction. $p$ values as indicated. Source data are provided as a Source Data file.

$1.4 \pm 0.0$ to $1.3 \pm 0.0$, $P < 0.01$, $n = 5$–$9$ mice/group, Fig. 4d, e). Further, reduced edema formation in intestines in antibody-treated animals was associated with reduced bacterial translocation. While peritoneal lavage showed no difference between CLP groups (Supplementary Fig. 2), less CFUs/mL tissue homogenate were found in the liver and the blood ($22173 \pm 16188$ vs. $76289 \pm 20713$ in the liver, $P < 0.05$, $n = 12$ mice/group and $3775 \pm 1367$ vs. $46340 \pm 12304$ per mL blood, $P < 0.001$, $n = 12$ mice/group Fig. 4f, g). Since vasoplegia is a common clinical phenomenon in patients with systemic inflammation, we evaluated the response of murine mesenteric resistance vessels from septic or control-treated, sham-operated mice to the vasoconstrictor phenylephrine. Vessels from septic mice exhibited a reduced response to the vasopressor ($0.6 \pm 0.04$-fold vasoconstriction compared to that seen in control-treated animals defined as 1.0, $P < 0.05$, $n = 7$-$11$ mice/group). In contrast, septic mice that had received truncated procalcitonin-neutralizing antibodies exhibited reduced vasoplegia ($1.5 \pm 0.2$-fold, $P < 0.001$, Fig. 4h). Further, antibody-treated mice exhibited preserved bioavailability of nitric oxide (NO) of the endothelium by showing preserved response to vasodilation-inducing agents (response to carbamoylcholine, reduction to $73.7 \pm 4.6\%$ vaso-constriction vs. $98.3 \pm 0.8\%$ in control-treated septic animals, $P < 0.001$, $n = 5$–$7$, Fig. 4i, j).

**Procalcitonin neutralization protects organ integrity**
As a possible summation of the above described effects, we next evaluated measures of organ integrity in mice subjected to treatment with procalcitonin targeting antibodies with focus on liver and kidney. While sepsis induced severe damage to liver tissue and interstitial fluid accumulation, these adverse effects were diminished after application of procalcitonin-targeting antibodies ($1.4 \pm 0.3$ vs. $3.1 \pm 0.2$ hepatic tissue damage score, $P < 0.01$, $n = 9$-$11$ and $1.0 \pm 0.0$ vs. $1.1 \pm 0.0$ wet/dry ratio, $P < 0.001$, $n = 11$, Fig. 5a–c). Further, antibody administration diminished acute tubular injury in murine kidneys ($1.8 \pm 0.3$ with control-IgG vs. $3.2 \pm 0.2$ with antibody, $P < 0.01$, $n = 7$-$13$, Fig. 5d, e) and protected the integrity of the glomeruli ($P < 0.001$, $n = 4$-$9$, Fig. 5f-i). These results indicated that neutralizing procalcitonin with antibodies designed for targeting the truncated, active procalcitonin variant iso-form, is a potent measure for organ protection during sepsis.

**PCT neutralization improves murine health conditions**
To evaluate whether reduced endothelial activation, capillary leakage and improved organ integrity result improved clinical wellbeing and health, we scored the general health condition, murine behavior and sepsis-related disease criteria hourly during the course of sepsis. Scores varied between antibody-treated mice and control-IgG-treated mice and indicated improved wellbeing, health and reduced sepsis-related disease severity in mice subjected to procalcitonin neutralization ($2.55$ vs. $3.55$ at 14 h, $P = 0.024$ until $4.75$ vs. $6.6$ at 18 h, $P < 0.001$ for health score, $1.85$ vs. $2.85$ at 6 h, $P = 0.029$ until $5.65$ vs. $7.95$ at 18 h,

$P < 0.001$ at 18 h for behavior score and $5.45$ vs. $7$ at 15 h, $P = 0.04$ until $7.7$ vs. $12.6$ at 18 h, $P < 0.001$ for sepsis score and $18.3$ vs. $26.1$ for the cumulative scores at 18 h, $P < 0.001$, Fig. 6a–d). Based on an endpoint defined as 20 in either score category, we calculated predicted survival that showed a significant advantage for mice treated with procalcitonin-targeting antibodies. Of note, these effects of the pro-calcitonin antibody were not observed with inhibitors targeting the procalcitonin receptor complex (BIBN4096, Fig. 6e, f).

## Discussion
The endothelium makes up the inner lining of all blood vessels. Instead of constituting a mere anatomical layer, the endothelium is considered as one of the most important functional players involved in the reg-ulation of coagulation, perfusion and immunity[22]. Although the endothelial transcriptome has been characterized extensively using bulk[23] and single cell RNA sequencing approaches[24], only little is known on how the endothelium is subject to change during disease and the extent of efficacy of therapeutic approaches to return the endothelium towards the transcriptomic profile associated with the healthy condi-tion. Sepsis is defined as a dysregulated immune response affecting the hosts tissue integrity[2]. High concentrations of inflammatory mediators and activated immune cells come into contact with 4000–6000 m² of the endothelial monolayer and we here show sepsis induces the dif-ferential expression of more than ~4000 genes in the endothelium. Although the number of genes regulated up or down was almost similar, up-regulated pathways showed a clear involvement in pro-inflammatory responses, regulation of cytokine signaling, leukocyte migration, adhesion and immune responses. In contrast, down-regulated pathways were associated with cell growth, morphogen-esis and development, suggesting that sepsis deeply affects the endothelial cell transcriptome inducing massive changes to endothe-lial integrity and function. In line with this assumption, we further show that mice–at the time point of transcriptome analysis–exhibited severe capillary leakage in various organs, barrier dysfunction and vasoplegia, all hallmarks of patients with sepsis. All of these entities compromise tissue perfusion and we detected severe tissue damage in liver and kidney, with the latter exhibiting dysfunction in almost all patients suffering from severe sepsis. During the past decades, various thera-pies have been clinically evaluated and almost all of them failed to significantly improve patient outcomes, making the delivery of anti-biotics and fluids as well as vasopressors for the antagonization of hypovolemia and hypotension the first and only (apart from surgical focus control if possible) treatment regimens for patients with sepsis[2]. Given that the hallmarks of clinical presentation of patients with sepsis are all highly related to vascular pathology, the vasculature is clearly an identified target for novel therapeutic strategies[3].

Procalcitonin has long been used as a sepsis biomarker due to its rapid plasma kinetics during early stages of sepsis[9] Interestingly, we here found that the procalcitonin encoding gene, *Calca*, was also

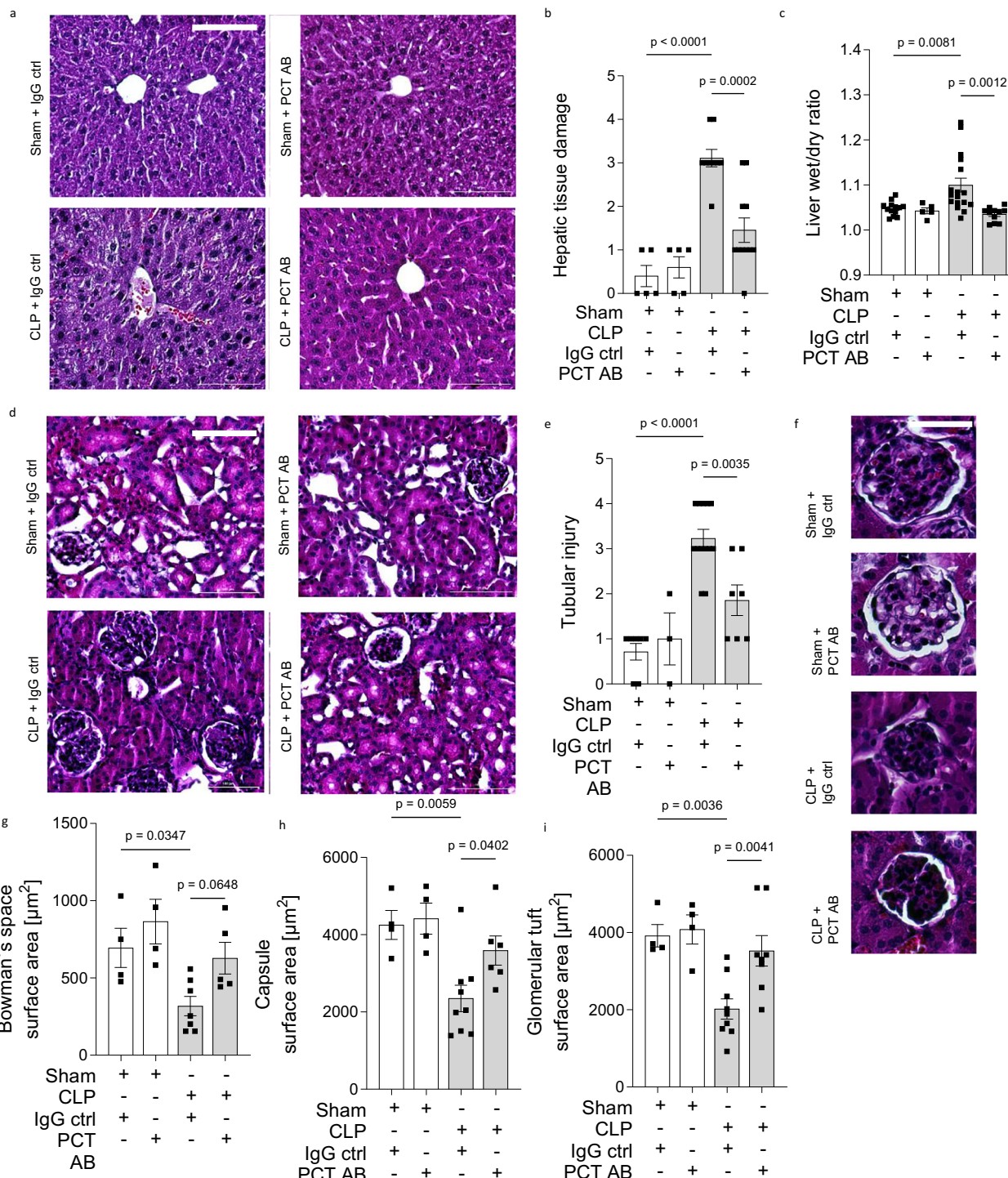

**Fig. 5 | Procalcitonin neutralization protects against organ damage during sepsis. a** Representative pictures showing hepatic tissue and **b** quantitative summary of hepatic tissue damage 18 h after CLP/sham surgery following antibody treatment. $n = 5$ (sham+ctrl, sham+AB), $n = 9$ (CLP+ctrl), $n = 11$ (CLP + AB) mice/group. Scale bar indicates 100μm. **c** Liver tissue edema of septic mice treated with the procalcitonin antibody or respective control IgG indicated as tissue wet/dry ratio 18 h after sepsis induction and 12 h after antibody treatment, $n = 5$(sham+AB), $n = 13$(sham+ctrl), $n = 11$(CLP + AB), $n = 17$ (CLP+ctrl) mice/group. **d** Representative pictures showing renal tissue and **e** quantitative summary of tubular injury 18 h after CLP/sham surgery and 12 h following antibody treatment, $n = 3$ (Sham+AB), $n = 7$ (Sham+ctrl, CLP + AB),$n = 13$ (CLP+ctrl) mice/group. Scale bar indicates 100

μm. **f** Representative pictures showing glomerular integrity and (**g–i**) quantitative summary of Bowman's space, capsule size and glomerular tuft surface area 18 h after CLP/sham surgery following antibody treatment, One-way ANOVA and Bonferroni-correction, multiple comparison of indicated pairs of colums. Bowmans space: $n = 4$(sham+ctrl, sham+AB), $n = 5$ (CLP + AB), $n = 7$ (CLP+ctrl), capsule surface: $n = 4$(sham+ctrl, sham+AB), $n = 68$(CLP + AB), $n = 9$ (CLP+ctrl), glomerular tuft: $n = 4$(sham+ctrl, sham+AB), $n = 5$ (CLP + AB), $n = 7$ (CLP+ctrl)mice/group. Scale bar indicates 50 μm. Data is presented as mean ± SEM. One-way ANOVA and Bonferroni-correction. *p* values as indicated. Source data are provided as a Source Data file.

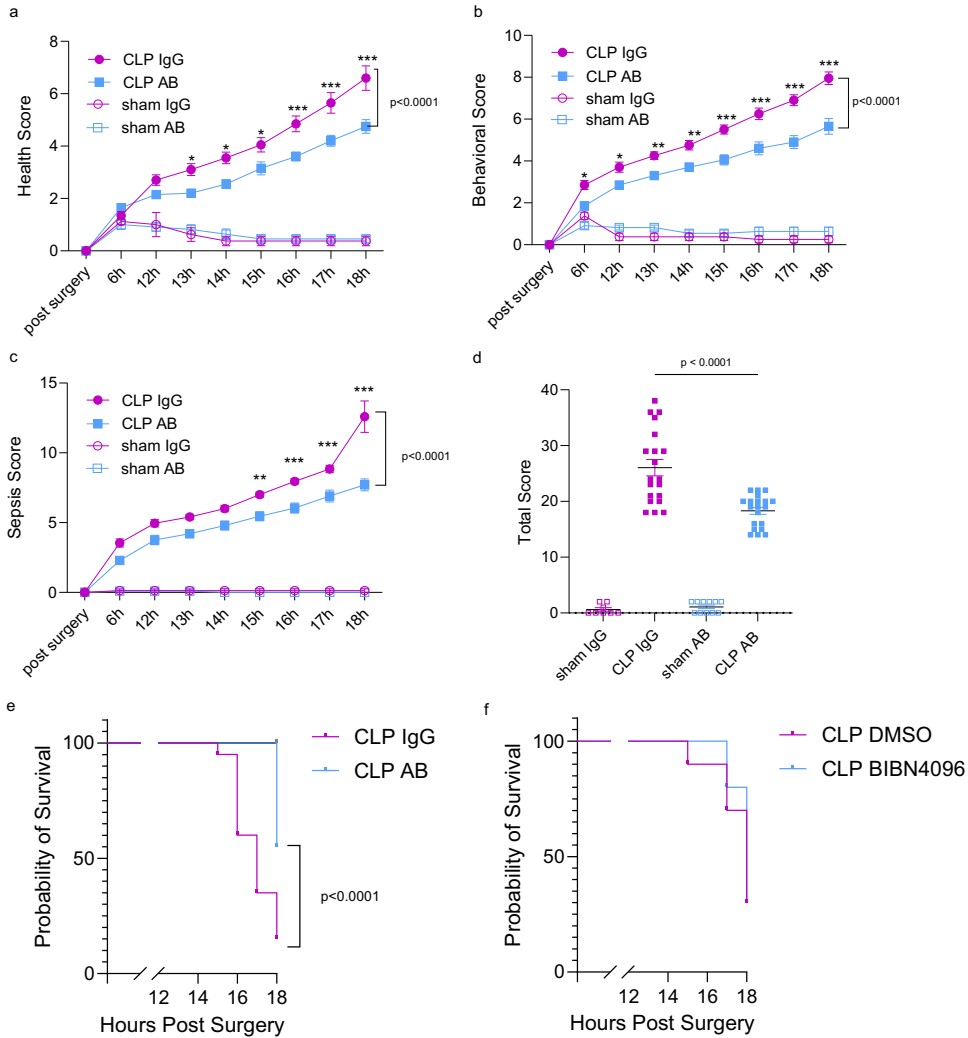

**Fig. 6 | Scores and predicted survival in mice treated with procalcitonin antibody. a–c** Health, behavior and sepsis-associated disease scores in septic or control mice treated with anti-procalcitonin antibody or control IgG ($n = 8$ (sham IgG), $n = 20$ (sepsis groups), $n = 11$ (sham+AB)). Data is presented as mean ± SEM. Two-way ANOVA and Bonferroni-correction. *$P < 0.05$, **$P < 0.01$, ***$P < 0.001$ vs. control. **d** Summation of score values at 18 h post sepsis induction, $n = 8$ (sham+IgG), $n = 11$ (sham+AB), $n = 20$ (CLP+IgG, CLP + AB), $p < 0.0001$. Data is presented as mean ± SEM. One-way ANOVA

and Bonferroni-correction. *$P < 0.05$, **$P < 0.01$, ***$P < 0.001$ **e** Kaplan-Meier-Curves showing probability of survival based on reach of the endpoint of at least 20 points in either score in septic mice treated with control IgG or procalcitonin antibody. $P < 0.0001$ vs. control, log rank (Mantel Cox) test, $n = 20$. **f** Similar scoring system in mice treated with the procalcitonin receptor inhibitor BIBN4096 or DMSO serving as vehicle control (no significant difference, log rank (Mantel Cox) test, $n = 10$). $p$ values as indicated. Source data are provided as a Source Data file.

significantly up-regulated during sepsis, suggesting that endothelial cells contribute to procalcitonin plasma concentrations during sepsis. Already two decades ago, it became clear that antagonizing procalcitonin during early stages of sepsis but also later during the course of disease augments macrohemodynamics and organ function in large animal models[25,26]. Also, antagonizing procalcitonin improved survival of septic rodents[27] and, in particular, neutralization of procalcitonins N-terminus protected against sepsis-induced acute lung injury[28]. Here, we show that a custom-made polyclonal antibody targeting the N-terminus of truncated, active procalcitonin can reduce the sepsis-induced changes to endothelial transcriptomic profiles. Antibody delivery resulted in cutting the number of both up and down-regulated genes in half. Importantly, this observation was associated with robust functional improvements of endothelial barrier function ameliorated capillary leakage in lungs and intestines, reducing bacterial translocation and vasoplegia, augmenting nitric oxide (NO) bioavailability and protecting septic animals from liver and kidney injury.

Apart from inhibiting the direct interaction of activated procalcitonin with the CRLR/RAMP1 receptor complex, additional mechanisms may underlie the beneficial effects of reducing

procalcitonin action during sepsis. For example, we and others have shown direct cell-damaging effects of procalcitonin on liver[29] and mesangial cells[30]. In addition, previous reports suggested procalcitonin mediates its direct contribution to sepsis pathology by inducing expression of interleukin-17A in gamma delta T-cells as immunoneutralization of IL-17A prevented from the deleterious effects of procalcitonin similarly as blockade of the procalcitonin receptor[31]. In endothelial transcriptomic profiles of septic mice, we found a significant induction of *Il17a* and *Il17f*, identifying endothelial cells as a previously unrecognized potential source of IL-17 in sepsis. IL-17A accelerates the release of chemokines and proinflammatory mediators from fibroblasts, endothelial cells and leukocytes[32]. In response to procalcitonin neutralizing antibody exposition but also receptor blockade and inhibition of procalcitonin activation, we observed a marked reduction of IL-17A in septic mice plasma, suggesting that inhibition of IL-17 signaling is an additional mechanisms of the beneficial effects of antagonizing procalcitonins action during sepsis.

Studies investigating transcriptomic profiles of the endothelium using single cell RNA sequencing have identified substantial heterogeneity among endothelial cells depending on their organ of origin

and their location within the vascular tree[24]. These studies also exhibited the largest population of homogenous endothelial profiles, i.e., associated with capillary localization, within the lungs. Taken together with the fact that the lung is one of the most susceptible organs to sepsis, we assume that our results are representative to the massive interferences of sepsis with endothelial homeostasis despite our use of a bulk sequencing approach and analysis of murine endothelial cells from only one organ. In fact, we here mirror the clinically relevant situation of massive changes to an organ remote to the sepsis focus, i.e., bacterial peritonitis. Procalcitonin is equally activated by N-terminal truncation in mice and humans and we have previously shown that inhibition of procalcitonin activation can augment micro- and macrovascular function in patients who develop systemic inflammation after surgery[10]. The results obtained here support the notion that antagonizing the procalcitonin signaling pathway is a promising strategy to preserve vascular function and organ integrity during sepsis and systemic inflammation. This approach would also display a precision medicine strategy effective in those identified with particular hyperprocalcitonemia.

## Methods

### Murine sepsis model
Animal experiments were approved by the governmental ethical board at the Animal Care and Use Committee of North Rhine Westphalia, Germany, 81-02.04.2021.A396. Female and male C57Bl/6 J mice (#000664, The Jackson Laboratory) were housed in the central animal facility under pathogen-free conditions with standard diet food and water ad libitum and in a 12 h light/dark cycle. The animal facility was maintained at $23 \pm 2\,°C$. 10-12 week-old mice were used for polymicrobial sepsis induction by cecal ligation and puncture performed by one experienced researcher to avoid technical variations (3–20 animal per group, depending on experiment). Briefly, laparotomy was performed following the ligation of the distal cecum (14 mm) and two punctures of the cecum using a 20 G cannula. After replacement of the cecum into the abdomen, the incision was closed. Mice in the sham operated control group underwent the same surgical procedure without ligation nor puncture of the cecum[10]. Following CLP surgery, the mice displayed lethargy and reduced mobility, showed no interest in their surroundings, and did not engage in social behaviors such as sniffing or grooming. These observations are consistent with the typical clinical signs of sepsis. During the course of sepsis, mice were scored for general condition (fur and outer aspect, mouse grimace scale, food and water intake), behavior (spontaneous movement, reactions) and criteria related to sepsis (outer aspect, indications for dehydration, diarrhea, shivering, vigilance) after surgery, 6 h, 12 h and every consecutive hour until 18 h. Each score ranged between 0 and 20 points. A score of 20 in one category was considered the humane endpoint used for calculation of predicted survival.

### Patient Recruitment
Between February and November 2022, we recruited 19 adult patients from the intensive care units of the University Hospital Münster who met the Sepsis-3 definition criteria and exhibited hyperprocalcitonemia (serum concentration > 0.5 ng/mL). The recruitment was based on defined criteria and was performed without any bias (study protocol: https://clinicaltrials.gov/study/NCT05703802). Written informed consent was obtained from patients or their legal representatives. The study received approval from the Ethics Committee of the University Hospital Münster (ID: 2019-494-f-S) and was registered on ClinicalTrials.gov (registration number: NCT05703802, principal investigator: S. Kintrup, MD). Patients were excluded if they had immunological disorders, were taking immunosuppressive agents other than hydrocortisone, had viral or fungal sepsis, or had a terminal pre-existing disease. The study population consisted of 19 participants with a mean age of $69.5 \pm 2.9$ years. Of these, 12 were male and 7 were female. The

mean body weight was $84.4 \pm 5.9$ kg, and the mean body mass index (BMI) was $28.2 \pm 1.7$ kg/m². The average body surface area (BSA) was $2.0 \pm 0.1$ m². Upon recruitment, 15 mL of serum were collected from each patient using two serum gel monovettes. The samples were centrifuged, and the serum was stored at $−80\,°C$ until analysis. Data on hemodynamics, laboratory parameters, and drug dosing on the day of blood sampling were retrieved from the patient data management system (Quantitative Sentinel, GE). SOFA score variables were assessed immediately after blood collection. Total PCT levels were measured in all patients using an electrochemiluminescence immunoassay on a cobas e801 analyzer (Roche Diagnostics GmbH) in the central laboratory of the University Hospital Münster.

### Isolation of endothelial cells and transcriptome analysis
For isolation of endothelial cells[33], murine lungs were excised 18 h after sepsis induction. Dissociation of lung tissue into single-cell suspensions for subsequent cell separations was performed using the Miltenyi lung dissociation kit and standard protocol (130-095-927, Miltenyi Biotec). Lungs were dissected and collected in a C-tube containing Enzyme A and D and Buffer S. The preinstalled lung dissociation protocol 37C_m_LDK_1 on the Miltenyi Octo Dissociator with heaters was used immediately for obtaining single cell culture. Endothelial cells were isolated from suspensions using MACS Technology (Miltenyi Biotec). Cells were resuspended in PEB (Phosphate Buffered Saline, 2 mM EDTA, 0.5% bovine serum albumine) buffer following depletion of CD45+ cells by using CD45-coupled microbeads. After 15 min of incubation, cells were placed on a LS column in a magnetic separator and rinsed with PEB buffer three times. The CD45- fraction was collected and CD31 antibody-coupled beads were used for magnetic sorting of endothelial cells to obtain a pure cell culture. For RNA sequencing, RNA was isolated immediately. For permeability analysis, cells were seeded on transwell inserts (0.4 µm pore size, Corning) and incubated for 96 h until confluence. Purity and viability of isolated pulmonary endothelial cells were verified by flow cytometry analysis (Supplementary Fig. 1, BD Accuri C6 Plus). Following RNA isolation using RNeasy Mini Kit (Quiagen), RNA was subjected to next generation sequencing. Sequencing was performed at the Core Facility Genomics of the Medical Faculty Münster. Quality and integrity of total RNA was controlled on Agilent Technologies 2100 Bioanalyzer (Agilent Technologies; Waldbronn, Germany). PolyA+RNA were purified from 100 ng total RNA using Poly(A) mRNA Magnetic Isolation module Kit (NEB E7490L, New England Biolabs). The RNA Sequencing library was prepared with NEBNext® Ultra™ II Directional RNA Library Prep Kit for Illumina® (New England Biolabs). The libraries were sequenced on Illumina NextSeq 2000 using NextSeq2000 P2 Reagent Kit (238 cycles, paired end reads 2 × 111bp) with an average of 26.7 M reads per RNA sample. Reads were quality checked with package FastQC (version 0.11.4), and then processed by two different pipelines resulting in Figs. 1 and 3. For data in Fig. 3 reads were trimmed using Trimgalore (version 0.4.4) with default settings. Reads were mapped to mouse genome annotation mm11 (ENSMBL Mus_musculus.GRCm39, release 104) using STAR (version 2.5.2b)[34] with default settings. Further analyses and visualizations of data was performed using the R software package (version 4.2.1)[35]. Mapped reads were counted using RsubRead (version 1.32.4)[36]. Raw counts were normalized and log2 transformed using function rlogTransformation from the DESeq2 package (version 1.16.1)[37] and from the limma package (version 3.58.1)[38] and an increment was added to the normalized values to make all values positive. For identification of differentially expressed genes (DEGs), DEseq2 was used with the model design = ~ group and then contrasting the groups. Differentially expressed genes (DEGs) were identified using a multiple-testing adjusted $p$ value of <0.05 and exhibiting more than a 2-fold (log2 = 1) difference in expression levels. For Fig. 1: Reads were aligned, and quantified using integrated counting in the STAR package. Differential gene expression analysis was then performed using the limma

package (version 3.58.1)[38]. As the data for Fig. 1 originated from two batches of sequencing a model design = ~group+batch was applied. VENN diagrams were generated with the function vennPlot[39]. Functional analyses of DEGs were performed using the R software package cluster Profiler (version 4.10.1)[40]. Heatmaps were generated with the function heatmap2 of package gplots (version 3.1.3; https://github.com/talgalili/gplots) and package pheatmap (version 1.0.12.)[41]. KEGG pathway views were generated using package pathview, version 1.36.1[42]. We then related our findings to results previously obtained in a human system. Since there was no data set available of endothelial cells exposed to the septic condition in vivo, we correlated our genomic data to gene expression data set obtained from endothelial cells exposed to LPS. We searched the EndoDB database (https://endotheliomics.shinyapps.io/endodb/) with the term 'lung'. We obtained 29 hits, 13 were from human. Of these, 12 were from freshly isolated cells or primary cell cultures. No dataset was found for sepsis-treated lung endothelial cells. One micro-array dataset reported gene expression changes after treatment of primary endothelial cells with LPS (E-GEOD-5883) at 4 h, 8 h and 24 h post treatment and corresponding untreated controls, which we found appropriate for comparison. We downloaded the respective data from the GEO expression database (GSE5883[43]) and compared it our findings. Downloaded raw data was quantile normalized and log$_2$ transformed and differentially expressed probe sets (DEPs) were determined using the limma package[44,45] with the model.matrix( ~ 0 + group), and then contrasting groups. DEPs were identified using a multiple-testing adjusted $p$ value of <0.05 and exhibiting more than a 2-fold ($|\log 2| > 1$) difference in expression levels. 643 probe sets were differentially expressed at 4 h (treatment versus respective control), 827 DEPs at 8 h, and 195 DEPs at 24 h. We chose the 8 h treatment for our comparison since it showed the strongest response.

## Culture of human and murine endothelial cells, permeability assay

Human pulmonary endothelial cells (PromoCell) were cultured in cell culture media and exposed to serum from human sepsis patients diluted 1:1 in serum-free media. Murine endothelial cells were isolated from murine lungs using magnetic-bead-conjugated CD31-antibodies and brought into culture. Following several additional purification steps, cells were exposed to plasma of septic mice (10% with complete cell culture media) or recombinant procalcitonin (Abcam) as indicated. Endothelial cells were incubated with 1 μg/mL of the antibody targeting truncated procalcitonin and respective IgG control for two h following stimulation with septic mouse plasma diluted 1:1 in serum-free cell culture media and application of 1 mg/mL 70 kDa fluorescein isothiocyanate (FITC-) labeled dextran (Sigma Aldrich). After two h, fluorescence was measured at 520 nm using a plate reader (Synergy Mx, Biotek) in samples taken from lower chambers.

## Western blot

Following stimulation of endothelial cells and lysing with homburg lysis buffer, protein concentration was determined using PierceTM BCA Protein Assay Kit (ThermoFisher). Protein loading was adjusted to 20 μg and sodium dodecyl sulfate polyacrylamide gel electrophoresis (SDS-PAGE) was performed for protein separation. Proteins were transferred to a polyvinylidenefluoride (PVDF) membrane and blocked with 3% bovine serum albumin in tris buffered saline with tween 20 following incubation with primary antibodies targeting phosphorylated VE-cadherin at tyrosine 685 (abcam, ab119785, 1:1000), total VE-cadherin (Santa Cruz Biotechnology, sc9989, 1:200) and actin (Thermofisher, MA5-15739, 1:10000) at 4 °C over night. Horseradish peroxidase (HRP)-linked secondary antibodies, anti-rabbit IgG or anti-mouse IgG (Cell Signaling, 7074S, 7076S,1:1000, 1:5000, respectively), were incubated at room temperature for 1 h. ECL$^{TM}$ Prime Western Blotting System (GE Healthcare RPN2232) was used for protein detection.

## Quantitative real-time PCR (qPCR)

Total RNA was isolated using the RNeasy Plus Mini Kit (Qiagen, 74134). Complementary DNA (cDNA) was synthesized using the High Capacity cDNA Reverse Transcription Kit (Applied Biosystems, 4374966). Gene expression was quantified by real-time PCR using TaqMan™ Fast Advanced Master Mix (Applied Biosystems, 4444557) and TaqMan™ Gene Expression Assays (Thermo Fisher Scientific). For mouse samples, the following assays were used: *Vegfa* (Thermofisher, Mm00437306_m1), *Vegfc* (Thermofisher, Mm00437310_m1), IL17a (Thermofisher, Mm00439618_m1) IL17f (Thermofisher, Mm00521423_m1) and *Gapdh* (Thermofisher, Mm99999915_g1). For human samples following human TaqMan™ Gene Expression Assays were used: *Calca* (Thermofisher, Hs01100741_m1), *Vegfa* (Thermofisher, Hs00900055_m1), *Calcrl* (Thermofisher, Hs00907738_m1), *Ramp1* (Thermofisher, Hs00195288_m1), and *Gapdh* (Thermofisher, Hs02786624_g1).

## Recombinant procalcitonin variant isoforms, procalcitonin plasma concentrations and in vivo interference with the procalcitonin signaling pathway

Procalcitonin variant isoforms were synthesized recombinantly in E. coli as previously described[10]:Procalcitonin (PCT) was PCR-amplified from Calca cDNA in a pPCT-ScriptAmpSK(+) vector (Dharmacon MHS6278-202856628) using Q5 polymerase (New England Biolabs). FLAG-tagged variants were generated with primers introducing an NcoI site at the 5′ end and a HindIII site plus His-tag at the 3′ end. PCR products were cloned into pET28a(+) using NcoI-HF and HindIII-HF, transformed into BL21 (C2987H) E. coli, and sequence-verified.

For expression, cultures were grown to OD600 ≈ 0.7 and induced with 500 μM IPTG for 4 h. Cells were lysed after centrifugation, and soluble fractions were purified on HIS-Select HF Nickel Affinity Gel (Sigma). Proteins were eluted with 50 mM NaH$_2$PO$_4$, 300 mM NaCl, 250 mM imidazole, and the FLAG tag was removed by enterokinase (4 U, 12 h).

Endogenous procalcitonin was detected by murine ELISA (Cloud-Clone). For inhibition of the procalcitonin pathway in vivo, 100 μg/kg olcegepant or 5 mg/kg sitagliptin were applied intravenously for CRLR/RAMP1 blockade or DPP4 inhibition[10].

## Generation of a polyclonal antibody targeting murine truncated procalcitonin

Polyclonal antisera against mouse procalcitonin were raised by immunizing rabbits with the peptide LRSILESSPGMATLSC (equal to the truncated procalcitonin N-terminus amino acid sequence) coupled to KLH. For affinity purification of procalcitonin-specific antibodies from rabbit sera the peptide used for immunization was coupled to SulfoLink resin (Thermo Fisher Scientific) (Supplementary Fig. 1). Procalcitonin targeting antibody was injected intravenously to 100-fold excess of the mean procalcitonin concentration detected in mice (10 ng/mL), i.e., 1 μg/mL of blood volume, or IgG control six h after sepsis induction.

## Cytokine multiplex analysis

Cytokine multiplex analysis in murine plasma withdrawn 18 h after sepsis induction was performed according to the manufacturer's instructions (Cytokine & Chemokine 36-Plex Mouse ProcartaPlexTM Panel 1 A including IL-17a, Thermo Fisher Scientific) to analyze humoral characteristics of the ongoing inflammation. The microplate was analyzed in a Luminex analysis device.

## Miles assay

10 mg/kg bodyweight Evans Blue (Sigma) were injected 17.5 h after sepsis induction by cecal ligation and puncture and sham surgery[46]. 30 min later, mice were sacrificed and murine lungs and blood were harvested. Lungs were homogenized and dissolved in formamide following incubation at 60 °C for 18 h to extract extravasated Evans blue

dye from tissue. Lungs were centrifuged and supernatant was used for further analysis. Absorbance of lung-supernatant and plasma was measured at both 620 nm and 740 nm using a spectrometer. The degree of albumin extravasation was calculated as quotient of Evans Blue content tissue to plasma. Wet/dry ratio indicating edema formation was analyzed in murine liver and intestines after 7 days of incubation at 37 °C and weight measurement on a precision balance.

## Assessment of bacterial colony-forming units
Agar plates (Carl Roth) were produced under sterile conditions. Peritoneal lavage was performed by injection of 1 mL Dulbecco's Phosphate Buffered Saline (DPBS). Murine lungs, liver, kidney and spleen were dissolved in 1 mL DPBS and homogenized under sterile conditions. 30 μl whole blood was diluted with equal amounts of thioglycolate. 50 μl of the dilution were plated on agar plates. 50 μl of the peritoneal lavage and 10 μl of the tissue homogenates were plated. Colony forming unit counting was performed after 24 h of incubation at 37 °C.

## Histology
Murine lungs, liver and kidney were removed, kept in 4% formaldehyde at 4 °C, embedded with parraffin and 6μm thick sections were stained with hematoxylin-eosin. Five representative pictures per mouse were taken using Lionheart FX microscope (BioTek, Software Gen5 Version 3.05) by a blinded observer. Alveolar septal thickening was analyzed by measurement of 10 representative septa using Image J software (Version 1.51r, NIH). Histopathological changes in murine liver was scored based on the criteria congestion, edema, infiltration of immune cells and necrosis[47]. Tubular injury of murine kidney was quantified by presence or absence of epithelial necrosis, loss of brush border, cast formation and tubular dilatation. Glomerular tuft, Bowmans space and capsule size were measured using the ImageJ freehand selection tool.

## Pressure myography
For the assessment of vasomotor function, third-order mesenteric resistance arteries were dissected 18 h after sepsis induction and pulled on glass cannulas in a vessel chamber (Living Systems Instrumentation) in calcium-free buffer (10 mM HEPES, 140 mM NaCl, 5 mM KCl, 1.2 mM MgCl2, 10 mM glucose, 1 mM EGTA)[48]. Experiments were performed in calcium-containing buffer (Ca-free buffer without EGTA including 2 mM CaCl2). After development of a stable myogenic tone at 80 mmHg, vessels were exposed to 10 nM $(R)$-$(-)$-phenylephrine hydrochloride (Phe, Tocris) for maximum constriction. Relaxation response to increasing concentrations of carbamoylcholine ($10^{-9}$ to $10^{-4}$M, Tocris) was assessed using digital video edge detection.

## Statistical analysis
Data are presented as mean values with standard error of mean. To test for normal distribution, the Shapiro Wilk-Test was used. Statistical significance was determined by using One- or Two-way analysis of variance (ANOVA) followed by correction for multiple testing applying the Bonferroni method. Effect was estimated by difference in means and 95% confidence interval and $P$ values ≤ 0.05 were considered to be statistically significant. Survival curves were analysed using log rank test. All analyzes were performed using Graph Pad Prism Software 7 and 10 (Graph Pad, USA) or SPSS Statistics 14.

## Reporting summary
Further information on research design is available in the Nature Portfolio Reporting Summary linked to this article.

## Data availability
The data that support the findings of this study are mostly shown in the figures (single data points). Source data are provided with this paper. RNA sequencing data that support the findings of this study have been deposited in GEO with the accession codes GSE244815 and GSE244943 (https://www.ncbi.nlm.nih.gov/geo/query/acc.cgi?acc=GSE244815, https://www.ncbi.nlm.nih.gov/ geo/query/acc.cgi?acc= GSE244943") Source data are provided with this paper.

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

## Acknowledgements
This work was supported by the Deutsche Forschungsgemeinschaft (German Research Foundation) and Clinical Research Unit CRU342, to NMW (WA3786).

## Author contributions
L.B. conducted experiments, analysed data, wrote and revised the manuscript. K.E.M.H., S.K., L.C., and P.B. conducted experiments and analyzed the data. A.N., K.S., S.W., R.A., V.R., and S.K. analyzed data and revised the manuscript. P.M., N.S. and D.V. revised the manuscript. N.M.W. analysed data, wrote and revised the manuscript.

## Funding

## Competing interests
The authors declare no competing interests.

## Additional information

**Peer review information** *Nature Communications* thanks Yuichi Hattori, Raymond Langley, and the other, anonymous, reviewer(s) for their con-tribution to the peer review of this work. A peer review file is available.

**Publisher's note** Springer Nature remains neutral with regard to jurisdic-tional claims in published maps and institutional affiliations.

