## [Transparent Peer Review file · Nature Communications]

Endothelial cell responses in sepsis are attenuated by targeting truncated procalcitonin

Corresponding Author: Professor Nana-Maria Wagner

Version 0:

Reviewer comments:

Reviewer #1

(Remarks to the Author)
NatComm Review

The present article investigates the effect of polymicrobial sepsis on endothelial cell responses in relation to procalcitonin signaling. In this regard, the authors had shown in a previous study that vascular leakage, vasoplegia and microvascular dysfunction are regulated by procalcitonin in sepsis. To induce acute sepsis, in the current study the authors subjected WT mice to the Cecal Ligation and Puncture (CLP) procedure. Subsequently, the authors conducted next-generation RNA sequencing on endothelial cells derived from murine lungs following the induction of polymicrobial sepsis, with the objective of assessing transcriptomic changes. The presented RNA sequencing data indicates the expression of numerous genes that are differentially expressed, including expression of the Calca gene, encoding procalcitonin, and Il17a.

The study is well executed and written. However, the authors' principal objective is to examine the role of procalcitonin during sepsis, a topic that has already been extensively investigated. A harmful role of procalcitonin in sepsis has been demonstrated in multiple studies in a variety of different organisms, including mice, rats, and pigs (e.g., Baranowsky et al., 2021; Nylén et al., 1998; Wagner et al., 2002; Tavares et al., 2014). Although the described antibodies in the current study appear to be of novel design, the studies referenced above have already explored the therapeutic potential of antagonizing or neutralizing procalcitonin. Unfortunately, in the current study the key experiment, namely survival rates in mice treated with the novel Procalcitonin antibodies, is not reported. Although the RNA seq data of vascular endothelial cells with/without anti-procalcitonin treatment are interesting, the functional impact of procalcitonin on vascular integrity during sepsis has been reported by the same group before (Brabenec 2022). Also, the proposed role of Il17A as a downstream mediator of procalcitonin has already been described in the study by Baranowsky et al. 2021. Together, the study provides only limited conceptual advance and novelty, and, in my opinion, is thus better suited for a more specialized journal

Nylén ES, Whang KL, Steinwald PM et al. Procalcitonin increases mortality and procalcitonin recognizing antiserum improves mortality in an experimental model of sepsis. *Crit Care Med* 1998; 26:1001–1006.

Wagner KE, Martinez JM, Vath SD et al. Early immunoneutralization of calcitonin precursors attenuates the adverse physiologic response to sepsis in pigs. *Crit Care Med* 2002; 30:2313–2321.

Tavares E, Maldonado R, Miñano FJ. Immunoneutralization of endogenous aminoprocalcitonin attenuates sepsis-induced acute lung injury and mortality in rats. *Am J Pathol* 2014; 184:3069-83.

Baranowsky A, Appelt J, Kleber C, Lange T, Ludewig P, Jahn D, Pandey P, Keller D, Rose T, Schetler D, Braumüller S, Huber-Lang M, Tsitsilonis S, Yorgan T, Frosch KH, Amling M, Schinke T, Keller J. Procalcitonin Exerts a Mediator Role in Septic Shock Through the Calcitonin Gene-Related Peptide Receptor. *Crit Care Med*. 2021 Jan 1;49(1):e41-e52.

Brabenec L, Müller M, Hellenthal KEM, Karsten OS, Pryvalov H, Otto M, Holthenrich A, Matos ALL, Weiss R, Kintrup S,

Reviewer #2

(Remarks to the Author)

Calcitonin is a polypeptide hormone, which is made up of 32 amino acids and is produced mainly by parafollicular cells (C cells) in the thyroid gland, secreted by cells of the glandular ultimobranchial bodies. The overall effect of calcitonin is to lower the concentration of calcium in the blood when it has risen to an above the normal level. Procalcitonin, a protein that consists of 116 amino acids, is the peptide precursor of calcitonin. Multiple studies have shown that procalcitonin is involved in the pathogenesis of infections, and that it can be a useful diagnostic marker for infections such as bacterial pneumonia, bacterial sepsis and septic shock, meningitis, infectious endocarditis, pancreatitis, and urinary tract infections. Secretion of procalcitonin is stimulated by IL-1 β and tumor necrosis factor- α in patients with bacterial infections and by interferon- γ in patients with viral infections. The authors previously reported that procalcitonin induced endothelial barrier disruption (Wagner et al. *Anesth Alalg* 2017). Furthermore, their recent study has shown that targeting the procalcitonin receptor during sepsis-induced hyperprocalcitonemia can significantly reduce capillary leakage (Brabenec et al. *Am J Respir Crit Care Med* 2022). In this study, the authors demonstrated that more than half of transcriptomic changes in endothelial cells were reduced by anti-procalcitonin antibodies and this was functionally associated with preserved vascular integrity in lungs and intestines, reduced sepsis-induced vasoplegia, preserved endothelial NO bioavailability, and organ integrity during sepsis. They also found that neutralizing procalcitonin exerted protection of pulmonary endothelial and epithelial barriers in mice with sepsis-induced sepsis. While this study includes interesting findings, I have a number of concerns for the authors to consider.

Major comments are summarized below.

- 1) The disruption of endothelial integrity and functioning plays a crucial role in the development of sepsis-associated organ dysfunction. In regards to this, there are a number of excellent review articles. In the Introduction, the authors should cite these reviews.
- 2) The authors describe that “so far, most therapeutic strategies failed to improve sepsis outcome (page 4, line 101). Without a doubt, despite advances in overall medical care, sepsis continues to be a complex clinical entity with an unacceptably high mortality rate. The authors should provide several appropriate references for this.
- 3) The authors describe that the lung is the most susceptible organ to sepsis-induced tissue injury (page12, line 276-277). The authors should provide an appropriate reference pointing out that the respiratory system is the most affected organ of the body and the lung dysfunction is the first step in the development of multiple organ failure in septic patients.
- 4) The gene encoding procalcitonin (Calca) was among the top up-regulated genes, while the gene encoding the procalcitonin receptor (Calcr1) was among the top down-regulated genes (page 12, line 283-285). It would be easy to understand if RT-PCR analysis showing gene expressions for Calca and Calcr1 in pulmonary endothelial cells from sham-operated and CLP mice is visually provided.
- 5) The authors found that pathways associated with cell growth, angiogenesis, and assembly of cellular junctions were down-regulated (Fig. 1G) (page12, line 288-289). In this regard, VEGF mRNA level was significantly down-regulated in lung tissues from mice with CLP-induced sepsis (Tomita et al. *Naunyn-Schmiedeberg's Arch Pharmacol* 2020). This report may be considered by the Authors.
- 6) Kaplan-Meier survival curves examining whether neutralizing procalcitonin can improve mice subjected to CLP should be provided.
- 7) In conclusion, the authors state that antagonizing the procalcitonin signaling pathway would display a precision medicine strategy effective in those identified with particular hyperprocalcitonemia. What percentage are septic patients with hyperprocalcitonemia occupy in the whole septic patients?

Reviewer #3

(Remarks to the Author)

The authors utilize multi-omic techniques and some well-designed murine animal model experiments to determine the impact of neutralizing a procalcitonin transcript isoform in a murine model of sepsis. The experiments are very well performed and the study is very elegant. However, as the authors state in the introduction, no therapeutic discoveries have been made that can treat the sepsis syndrome. This is primarily related to the fact that most studies have been performed in rodent models. Unfortunately, the findings in mice rarely translate in humans. Therefore, it is important for the authors to provide some evidence that a similar pathway related to procalcitonin inhibition improved endothelial function in a human model such as primary cell culture, lung organoid, etc.

1. As the authors stated – the primary benefit of neutralizing procalcitonin, was the impact on endothelial and epithelial barrier integrity. The experiments are well performed and described. However, one finding is that the inhibition of the procalcitonin isoform also improves intestinal barrier function. Their results demonstrate that there are decreased CFU's in all organ tissues they looked at. It seems to me that a potential rationale for the reduced CFU's throughout the periphery could be related to the improved integrity of intestinal barrier function. Simply put, the bugs have less ability to cross past the gut therefore many of the inflammatory responses are tamped down. Clarification on this point is important to understand the impact of the potential therapeutic - is it improved endothelial barrier function in lung tissues, or the reduced bacterial load because the bugs can't cross?

2. In the methods, the authors state that human endothelial cell cultures were performed for permeability assays. However, the methods and results do not include any information about the human endothelial culture. This is unfortunate, as a human endothelial model that recapitulates the mouse findings would tremendously improve the impact of this study. Considering that the study utilizes plasma of septic mice, plasma from septic patients on human endothelial cell cultures could similarly be performed and would greatly enhance this study.

Similarly, it would be great if the authors could perform a similar RNAseq analysis, or at least target 5 -10 of the key genes identified in the mouse study, using the primary airway endothelial cell model and plasma challenge.

3. The recombinant procalcitonin variant study: This is an excellent component of the study and suggests that antagonizing this isoform could have potential therapeutic benefit. That said, there are a couple of areas that need to be clarified. First, in the methods the authors refer to the transcript isoforms as variants. I am not certain this is the correct terminology as that has me thinking about SNP variants. It would probably be easier for the reader to follow if you refer to it as a splice isoform or transcript isoform consistently throughout the manuscript.

More importantly, though, how does this isoform relate to human biology? I assume this isoform is also found in humans. A description of the human isoform would be helpful – how evolutionarily conserved are these isoforms in mice to humans? Evidence that the isoform can be found in human endothelial cells is necessary to demonstrate that the pathway is conserved.

Does the polyclonal antibody that was developed also antagonize the human isoform and have similar results? It would be helpful to know if the antibody developed has cross reactivity or if a new antibody needs to be developed.

The authors also discuss olcegepant and inhibition of DPP4 as abolishing the IL-17 mediated response. Anything that can confirm these results in a human model would be helpful to understand the potential translational impact of inhibiting this transcriptional isoform.

Minor concerns – The paragraphs are overly long. Particularly in the discussion. Breaking these up would make the manuscript easier to read.

Line 417 – custom is spelled incorrectly.

Reviewer #4

(Remarks to the Author)

Brabenec et al. performed bulk RNA sequencing of lung endothelial cells isolated from a mouse model of sepsis and identified transcriptomic changes that describe the impact of sepsis pathology on this critical population. This team has previously reported that a truncated form of procalcitonin disrupts vascular integrity using in vitro and in vivo studies that were supported with human clinical data from subjects that included those taking inhibitors of the procalcitonin activator DPP4. In new data reported here, the authors use a procalcitonin neutralizing antibody that was generated to test if inhibition blocks changes in the endothelial transcriptome as well as accompanying vascular defects and organ damage. This work addresses an important area of investigation. New host-based therapies are needed to improve sepsis outcomes. Beyond supportive care and antimicrobials there are few options available for clinicians to treat sepsis and septic shock. Below are some comments for clarifications to strengthen the paper prior to recommendation for publication.

The authors previously reported a role for procalcitonin in vascular pathology using a variety of experimental and observational approaches. This included mouse models of sepsis and various inhibition strategies to prove that procalcitonin is more than just a biomarker. In this study, the key new results are the transcriptomic evaluation of the lung endothelial population with respect to procalcitonin and expanded assessment of the protective role of procalcitonin inhibition on organ systems beyond the lung.

1. Because the transcriptomic analysis is a proposed major contribution of this study, there remain some questions that would be helpful to fully understand the data. The authors use enzymatic dissociation of the lung followed by CD45+ depletion and CD31 enrichment to generate "pure" endothelial populations. There are a number of detailed protocols (Conchinha NV 2021) that describe this procedure. Can the authors provide more detail in the methods on their protocol since it is provided "in brief". Similarly, is there data available to show that the isolated population is pure?

2. Also, in the methods there is a 96h expansion of the isolated endothelial cells. Is the RNAseq performed after this 96h expansion? It is not clear as written especially since the main text on line 278 indicates that the RNAseq was done 18h after

cecal puncture. This is an important detail since in vitro expansion of endothelial cells can cause phenotypic drift as measured by transcriptomics (Cleuren ACA 2019).

3. The generation of the antibody against the activated procalcitonin is used as a tool to neutralize the effects of procalcitonin on the endothelium. Although information is supplied in the supplement on the purified antibody using surface coating and data is shown that the truncated procalcitonin can be blocked experimentally, do the authors have data showing the specificity of the antibody? Either following overexpression and by Western blot showing the full blot or dose dependent inhibition in vitro of tyrosine phosphorylation on cadherin?

4. On line 319 the authors claim that the procalcitonin antibody “reduced the expression of almost all DEGs during sepsis”. On line 419 the authors indicate that the antibody cut the number of genes in half. Figure 3D shows significant heterogeneity across the individual animals tested. Can the authors clarify the interpretation of how this is described and discussed? This is a key point since this can be interpreted that activated procalcitonin is the primary driver of lung endothelial pathology at the genomic level. This data could be strengthened by assessing impacts on endothelial transcriptomes following a complementary inhibition such as those shown in 3F for IL-17a protein.

5. The authors and other groups have previously shown a causal link or correlation with vascular pathology and procalcitonin in multiple systems and models including sepsis. Therefore the gene expression analysis following procalcitonin seems to be the major new contribution here and could be a valuable dataset to understand therapeutic intervention in sepsis and translational potential. This data modality could also be a bridge to linking the CLP model of sepsis to any human pathology. Therefore, a major deficit of the study and a missed opportunity is the lack of any transcriptomic comparison of the data generated here to publicly available human datasets. It would be valuable to include an analysis of the genes regulated by procalcitonin inhibition to those human paradigms (endoDB). In addition, although the authors acknowledge in the discussion that single cell RNAseq of lung has shown that it is one of the least heterogeneous organs with respect to endothelial cell diversity (Kalucka 2020), this study would be much more impactful if a single cell analysis was performed to understand the action of procalcitonin on the endothelium. If this is out of scope for this study, it would be important to discuss how some of the specific genes regulated in this study compare or align to those specific to endothelial subtypes in Kalucka 2020.

Minor points:

On line 276 is there a reference that can be added to support the point that the “lung is the most susceptible organ” to sepsis induced injury?

In Figure 2E in column 4 of the Western Blot, is it correct that it is “trunc-PCT” alone, or is this with IgG as well?

Version 1:

Reviewer comments:

Reviewer #1

(Remarks to the Author)

The authors have revised the manuscript and figures, and included additional experimental data, in an effort to enhance the overall quality, clarity and translational value of the article under review. These efforts are recognised and appreciated. Nevertheless, I believe that my primary and most substantial concerns regarding the manuscript have largely gone unaddressed. The role of procalcitonin in the pathophysiology of sepsis has been extensively characterised in the literature, particularly with regard to the inhibition of Calcr1-signalling by the antagonist BIBN4096, which has been shown to improve survival in mouse models. Furthermore, it has been convincingly demonstrated that the observed survival benefit of Calcr1-signalling antagonism is mechanistically mediated via the induction of IL-17A, as reported by Baranowsky et al. (2021). It is important to note that while this proposed therapeutic intervention was found to be promising in small animal models, it failed translation in a more clinically relevant large animal model of porcine polymicrobial sepsis, as documented by Messerer et al. (2022). Consequently, the novel, clinically applicable information that is provided by the current manuscript remains unclear, especially due to the absence of a clearly defined clinical outcome measure. If the authors wanted to claim that their study provides meaningful novel insight of clinical relevance, they would have to provide compelling comparative data demonstrating that the truncated version of their novel procalcitonin antibody is significantly superior to the known CALCR antagonist BIBN4096. In the absence of such evidence, it is challenging to substantiate the claim of novelty or clinical translatability. In summary, I believe that the present manuscript makes only a negligible contribution to the current understanding of the role of procalcitonin in sepsis and falls short of providing substantial translational or clinical value and should therefore be published in a more specialized journal

Reviewer #2

(Remarks to the Author)

I appreciate that the authors have made a diligent revision of the manuscript. In the revised version, the authors have addressed all my comments appropriately and further enhanced the manuscript.

The manuscript has been improved at a reasonably significant level.

Reviewer #3

(Remarks to the Author)

The authors have addressed all my comments.

Reviewer #4

(Remarks to the Author)

R4.1: Thank you for the added data.

R4.2: Thank you for the clarification

R4.3: Thank you for the added data

R4.4: Thank you for the clarification

R4.5: I thank the authors for the addition of the human transcriptomics analysis.

R4.6: Thank you for the added references.

R4.7: Thank you for the correction.

Minor Comments:

I have noted a number of spelling corrections needed in the text and figure legends. Please do a final check.

The addition of the human cohort and human in vitro studies strengthens the manuscript. Please ensure that references to plasma and serum are used correctly throughout the report. The legends and methods indicate that serum was collected from the human subjects, but it seems that in the discussion of the results you indicate that plasma was used. Please double check. Also, can you indicate in the methods more specifically how experiments were performed with mouse or human serum (plasma) with respect to percent of human or mouse serum added to the cell culture media? Was FBS still included in these experiments?

Version 2:

Reviewer comments:

Reviewer #1

(Remarks to the Author)

Reviewer #2

(Remarks to the Author)

The manuscript has been strengthened by the Kaplan-Meier survival curves showing that neutralizing procalcitonin, but not BIBN4096, was able to improve mice after CLP surgery. Please mention that log rank test or Gehan-Breslow-Wilcoxon test was used for comparing Kaplan-Meier survival curves in the Statistical analysis section or in the legend of Fig. 6.

Version 3:

Reviewer comments:

Reviewer #2

(Remarks to the Author)

All the points raised by this reviewer have been addressed satisfactorily.

Point-by-Point-Response

REVIEWER COMMENTS

Dear Reviewers,

we very much appreciate the opportunity to substantially revise our manuscript and, importantly, extent its translational relevance and mechanistic insights. In response to your comments, we used septic patients plasma to dissect the effects on gene expression profiles of human endothelial cells and further specify the importance of the procalcitonin antibody mechanistically at the molecular level. We also correlated the genomic data obtained in the vasculature of septic mice with publicly available human datasets exposed to sepsis-like conditions to verify the translational relevance of our findings to the human system. Manuscript and figures were substantially revised and the majority of your comments addressed by additional experiments added to the revised version of the manuscript.

Based on the extensive experiments conducted for the manuscript, we gathered additional expertise and added two additional authors on the revised version of the manuscript. All authors have read and approved the final version of the manuscript.

We regard our findings of relevance to a broad readership of physicians, basic scientists and clinician scientists. We would thus appreciate a re-evaluation of our work for publication in *Nature Communication*.

Reviewer #1 (Remarks to the Author):

C1: The present article investigates the effect of polymicrobial sepsis on endothelial cell responses in relation to procalcitonin signaling. In this regard, the authors had shown in a previous study that vascular leakage, vasoplegia and microvascular dysfunction are regulated by procalcitonin in sepsis. To induce acute sepsis, in the current study the authors subjected WT mice to the Cecal Ligation and Puncture (CLP) procedure. Subsequently, the authors conducted next-generation RNA sequencing on endothelial cells derived from murine lungs following the induction of polymicrobial sepsis, with the objective of assessing transcriptomic changes. The presented RNA sequencing data indicates the expression of numerous genes that are differentially expressed, including expression of the Calca gene, encoding procalcitonin, and Il17a.

The study is well executed and written. However, the authors' principal objective is to examine the role of procalcitonin during sepsis, a topic that has already been extensively investigated. A harmful role of procalcitonin in sepsis has been demonstrated in multiple studies in a variety of different organisms, including mice, rats, and pigs (e.g., Baranowsky et al., 2021; Nylén et al., 1998; Wagner et al., 2002; Tavares et al., 2014). Although the described antibodies in the current study appear to

be of novel design, the studies referenced above have already explored the therapeutic potential of antagonizing or neutralizing procalcitonin. Unfortunately, in the current study the key experiment, namely survival rates in mice treated with the novel Procalcitonin antibodies, is not reported.

Although the RNA seq data of vascular endothelial cells with/without anti-procalcitonin treatment are interesting, the functional impact of procalcitonin on vascular integrity during sepsis has been reported by the same group before (Brabenec 2022). Also, the proposed role of IL17A as a downstream mediator of procalcitonin has already been described in the study by Baranowsky et al. 2021. Together, the study provides only limited conceptual advance and novelty, and, in my opinion, is thus better suited for a more specialized journal.

R1->C1. We thank the reviewer for his remarks. The goal of translational research advances is to identify novel *therapeutic* strategies for the benefit of patients. This is of interest to both basic scientists as well as clinicians. Sepsis is a disease with high incidence and high and unchanged mortality rates over the past decades, making sepsis a substantial burden to the health care system and society. The organ dysfunction induced by sepsis is undisputedly connected to a loss of vascular integrity (for example, Joffe J et al., Am J Resp Crit Care Med, 2020). However and although urgently needed and a promising therapeutic opportunity, the impact of therapeutic strategies targeting the endothelium in sepsis has not been systematically dissected so far.

We here present the first endothelial transcriptome in the clinically most relevant disease model of a polymicrobial sepsis at a time point where most patients with sepsis are diagnosed. The transcriptomic data provides extensive insights in the pathways up- and down-regulated in the vasculature at a time point, where mice exhibit sepsis symptoms and elevated levels of procalcitonin. Procalcitonin is one of the most widely used biomarkers in sepsis, however, we were the first to describe a molecular pathway induced by activated procalcitonin that directly mediates loss of vascular barrier function, the most critical hallmark in the pathophysiology of septic patients. However, the verification of the use of targeting procalcitonin actions on the endothelium was missing.

We here verify antagonizing procalcitonin massively cuts down the number of regulated genes in the septic endothelium, ameliorates loss of vascular barrier function which results in robust organ protection during sepsis. We thus assume our results of relevance to a wide audience, including basic science researcher and clinician scientist as well as physicians and thus the broad audience of Nature Communications.

Unfortunately, survival experiments are not in accordance with the local authorities. However, we present robust data showing that organ integrity is preserved with regards to various aspects in organs most affected by lethal sepsis such as the lung, the kidney and the liver.

Reviewer #2 (Remarks to the Author):

C2.0: Calcitonin is a polypeptide hormone, which is made up of 32 amino acids and is produced mainly by parafollicular cells (C cells) in the thyroid gland, secreted by cells of the glandular ultimobranchial bodies. The overall effect of calcitonin is to lower the concentration of calcium in the blood when it has risen to an above the normal level. Procalcitonin, a protein that consists of 116 amino acids, is the peptide precursor of calcitonin. Multiple studies have shown that procalcitonin is involved in the pathogenesis of infections, and that it can be a useful diagnostic marker for infections such as bacterial pneumonia, bacterial sepsis and septic shock, meningitis, infectious endocarditis, pancreatitis, and urinary tract infections. Secretion of procalcitonin is stimulated by IL-1 β and tumor necrosis factor- α in patients with bacterial infections and by interferon- γ in patients with viral infections. The authors previously reported that procalcitonin induced endothelial barrier disruption (Wagner et al. Anesth Alalg 2017). Furthermore, their recent study has shown that targeting the procalcitonin receptor during sepsis-induced hyperprocalcitonemia can significantly reduce capillary leakage (Brabenec et al. Am J Respir Crit Care Med 2022).

In this study, the authors demonstrated that more than half of transcriptomic changes in endothelial cells were reduced by anti-procalcitonin antibodies and this was functionally associated with preserved vascular integrity in lungs and intestines, reduced sepsis-induced vasoplegia, preserved endothelial NO bioavailability, and organ integrity during sepsis. They also found that neutralizing procalcitonin exerted protection of pulmonary endothelial and epithelial barriers in mice with sepsis-induced sepsis. While this study includes interesting findings, I have a number of concerns for the authors to consider. Major comments are summarized below.

R2.0: We thank the Reviewer for his important remarks.

C2.1: The disruption of endothelial integrity and functioning plays a crucial role in the development of sepsis-associated organ dysfunction. In regards to this, there are a number of excellent review articles. In the Introduction, the authors should cite these reviews.

R2.1: We appreciate the reviewer's insightful comment. We fully agree that the loss of endothelial cell integrity and function represents a critical hallmark in the pathogenesis of sepsis-associated organ dysfunction. To further emphasize this point, we have incorporated additional references into our manuscript (page 3) that highlight the role of endothelial dysfunction in sepsis (reference No. 5, 6 and 7 of the revised manuscript):

- WL Lee, AS Slutsky. Sepsis and endothelial permeability, N Engl J Med, 2010

- Ince, Can; Mayeux, Philip R.; Nguyen, Trung; Gomez, Hernando; Kellum, John A.; Ospina-Tascón, Gustavo A.; Hernandez, Glenn; Murray, Patrick; De Backer, Daniel on behalf of the ADQI XIV Workgroup. The Endothelium in Sepsis. *SHOCK* 45(3):p 259-270, March 2016. | DOI: 10.1097/SHK.0000000000000473
- Joffre J, Hellman J, Ince C, Ait-Oufella H. Endothelial Responses in Sepsis. *Am J Respir Crit Care Med*. 2020;202(3):361-370. doi:10.1164/rccm.201910-1911TR

These reviews provide comprehensive insights into the mechanisms of endothelial dysfunction in sepsis and reinforce its pivotal role in organ failure.

C2.2) The authors describe that “so far, most therapeutic strategies failed to improve sepsis outcome (page 4, line 101). Without a doubt, despite advances in overall medical care, sepsis continues to be a complex clinical entity with an unacceptably high mortality rate. The authors should provide several appropriate references for this.

R2.2: According to the reviewers suggestion, we added several examples of studies that showed that most targeted approaches in sepsis failed (i.e. antagonizing IL-1, TNFalpha or application of hydrocortison (references No. 15, 16, and 17 of the revised manuscript).

C2.3) The authors describe that the lung is the most susceptible organ to sepsis-induced tissue injury (page12, line 276-277). The authors should provide an appropriate reference pointing out that the respiratory system is the most affected organ of the body and the lung dysfunction is the first step in the development of multiple organ failure in septic patients.

R2.3: We appreciate the reviewer’s comment and added a supporting reference that has dissected the massive sepsis-induced destruction on the lungs (References No. 34-36).

C2.4) The gene encoding procalcitonin (*Calca*) was among the top up-regulated genes, while the gene encoding the procalcitonin receptor (*Calcr1*) was among the top down-regulated genes (page 12, line 283-285). It would be easy to understand if RT-PCR analysis showing gene expressions for *Calca* and *Calcr1* in pulmonary endothelial cells from sham-operated and CLP mice is visually provided.

R2.4: We thank the reviewer for this important point. To clarify the dynamics of *Calca* expression in endothelial cells during sepsis and add to the translational importance of the data, we exposed human pulmonary microvascular endothelial cells to plasma from septic patients exhibiting procalcitonin levels between 4 and 70ng/mL. Here, we could show a robust induction of *Calca*. Importantly, this effect was abolished by procalcitonin antibodies, indicating that procalcitonin itself can perpetuate its expression and that endothelial cells are a source of procalcitonin during sepsis. Instead, we exhibited no regulation of the procalcitonin receptor encoding gene *Calcr1*. Interestingly, we also observed a trend towards a downregulation of the receptor activity modifying protein 1 encoding gene (*Ramp1*) in human endothelial cells exposed to septic patients plasma. RAMP1 is required for the CRLR to function as the procalcitonin receptor. We included this data in Figure 1 of the revised version of the manuscript.

C2.5) The authors found that pathways associated with cell growth, angiogenesis, and assembly of cellular junctions were down-regulated (Fig. 1G) (page12, line 288-289). In this regard, VEGF mRNA level was significantly down-regulated in lung tissues from mice with CLP-induced sepsis (Tomita et al. Naunyn-Schmiedeberg's Arch Pharmacol 2020). This report may be considered by the Authors.

R2.5: We thank the reviewer for this important point. To strengthen our results comparing it to previously published reports and to add to the translational relevance of the data, we analyzed VEGF mRNA expression in human pulmonary endothelial cells exposed to plasma from septic patients. We exhibited a downregulation of VEGF mRNA under exposition with septic patients plasma (Figure 1L).

We also subjected mice to procalcitonin injection and analyzed endothelial gene expression in pulmonary endothelial cells isolated after 18h. We here also found PCT induces a trend towards a reduction in VEGFa and c mRNA expression (Figure 1H). We also included the report from Tomita and colleagues as a reference into our manuscript (reference number 37).

C2.6) Kaplan-Meier survival curves examining whether neutralizing procalcitonin can improve mice subjected to CLP should be provided.

R2.6: Very unfortunately, the local authorities do not permit survival experiments.

C2.7) In conclusion, the authors state that antagonizing the procalcitonin signaling pathway would display a precision medicine strategy effective in those identified with particular hyperprocalcitonemia. What percentage are septic patients with hyperprocalcitonemia occupy in the whole septic patients?

R2.7: We thank the reviewer for his remark. Approximately 80% of all septic patients exhibit hyperprocalcitonemia (i.e. above 0.5ng/mL concentration in serum).

Reviewer #3 (Remarks to the Author):

C3.0: The authors utilize multi-omic techniques and some well-designed murine animal model experiments to determine the impact of neutralizing a procalcitonin transcript isoform in a murine model of sepsis. The experiments are very well performed and the study is very elegant. However, as the authors state in the introduction, no therapeutic discoveries have been made that can treat the sepsis syndrome. This is primarily related to the fact that most studies have been performed in rodent models. Unfortunately, the findings in mice rarely translate in humans. Therefore, it is important for the authors to provide some evidence that a similar pathway related to procalcitonin inhibition improved endothelial function in a human model such as primary cell culture, lung organoid, etc.

R3.0: We sincerely thank the reviewer for the positive remarks and acknowledge that many previous attempts to translate findings into the human model have been unsuccessful. However, in a prior study, we investigated the effects of inhibiting the procalcitonin pathway in patients undergoing on-pump cardiac surgery, a population known to exhibit elevated procalcitonin levels postoperatively, which correlate with capillary leakage, microcirculatory dysfunction and poorer outcomes.

Our findings demonstrated that cardiac surgery patients receiving sitagliptin, a DPP4 inhibitor that suppresses procalcitonin activation, required less intraoperative fluid administration and norepinephrine to maintain comparable mean arterial pressure—suggesting improved vascular integrity. Additionally, these patients exhibited a higher proportion of perfused vessels and an increased microvascular flow index, both indicative of improved microcirculatory function and overall outcomes (Brabenec L et al., Targeting procalcitonin protects vascular barrier integrity, *Am J Res Crit Care Med* 2022, <https://doi.org/10.1164/rccm.202201-0054OC>).

[REDACTED]

These findings underline the potential of targeting the procalcitonin pathway in a clinical setting, highlighting its promise as a therapeutic approach in sepsis—not only in animal models but also in human patients (Brabenec et al., Am J Resp Crit Care Med, 2022)

C3.1. As the authors stated – the primary benefit of neutralizing procalcitonin, was the impact on endothelial and epithelial barrier integrity. The experiments are well performed and described. However, one finding is that the inhibition of the procalcitonin isoform also improves intestinal barrier function. Their results demonstrate that there are decreased CFU's in all organ tissues they looked at. It seems to me that a potential rational for the reduced CFU's throughout the periphery could be related to the improved integrity of intestinal barrier function. Simply put, the bugs have less ability to cross past the gut therefore many of the inflammatory responses are tamped down. Clarification on this point is important to understand the impact of the potential therapeutic - is it improved endothelial barrier function in lung tissues, or the reduced bacterial load because the bugs can't cross?

R3.1 Our hypothesis is, that procalcitonin directly affects barrier stability. According to the reviewers suggestions, we now began to dissected the role of procalcitonin on endothelial vs. epithelial barrier integrity and found that – contrary to the endothelial barrier – procalcitonin did not affect epithelial integrity in an in vitro assay of intestinal epithelial cells. These evaluations strengthen the notion that procalcitonins action is

specifically exerted on the endothelium and that this is crucial for the reduction in bacterial translocation during sepsis.

Using Caco-2 cells in a two-chamber transwell system, the trans-epithelial resistance (TER) through the cell monolayer was measured to evaluate monolayer integrity. Six hours after exposure, TER was similar in the PCT, Sita and PCT+Sita groups and did not differ from the control group (DMEM + 10% FBS). These findings show that none of the substance(s) alters the integrity of the Caco-2 monolayer *in vitro* and suggest that PCT does not affect gut epithelial barriers.

Figure 2: Caco-2 cells barrier integrity in response to exposure of procalcitonin. TER-values across Caco-2 cells seeded on transwell inserts. TER values after 6 h incubation of 10 ng/ml PCT, 1 μM Sita or PCT+Sita did not differ significantly from control (DMEM + 10% FBS) and between groups. $N = 4$, $n = 29$ PCT, $n = 28$ PCT+Sita, $n = 25$ Sita, One-way ANOVA, Tukey post hoc analysis, ns = not significant.

C3.2.1 In the methods, the authors state that human endothelial cell cultures were performed for permeability assays. However, the methods and results do not include any information about the human endothelial culture. This is unfortunate, as a human endothelial model that recapitulates the mouse findings would tremendously improve the impact of this study. Considering that the study utilizes plasma of septic mice, plasma from septic patients on human endothelial cell cultures could similarly be performed and would greatly enhance this study.

R3.2.1: We thank the reviewer for this comment. In a prior study, that included human data (please also see answer 3.0), we investigated the effects of human procalcitonin and the inhibition of the procalcitonin pathway in human endothelial cells. In this study we were able to demonstrate that procalcitonin is the driver of loss of endothelial barrier integrity (Brabenec L et al., Targeting procalcitonin protects vascular barrier integrity, *Am J Resp Crit Care Med*, 2022, <https://doi.org/10.1164/rccm.202201-0054OC>). We also showed that patients with hyperprocalcitonemia in the setting of postoperative sterile systemic inflammation exhibit capillary leakage, microcirculatory dysfunction and increased fluid demands due to hypovolemia.

In response to the reviewers comment, we now extended this data further by including data on gene expression exerted by procalcitonin in human endothelial cells as well as a newly conducted biostatistical analysis correlating our findings in the murine septic endothelium to a human dataset obtained from EndoDB in Figure 1M (exposure of human endothelial cells to LPS) in the revised manuscript. In addition, we also

conducted experiments analyzing septic patients plasmas effect on human endothelial cells and show that procalcitonin antagonization (with a polyclonal antibody specifically targeting human truncated procalcitonin) can specifically reduce septic patients plasma molecular effects on endothelial barrier disruption in human endothelial cells:

Figure 3: A,B VE-Cadherin phosphorylation is increased in human endothelial cells upon stimulation with septic patients serum. C Increased endothelial cell permeability after septic human serum stimulation is reduced by an antibody targeting procalcitonins N-terminus that resulted in a reduction of VE-Cadherin phosphorylation in response to septic patients plasma (D, E). F Schematic overview.

These results are now provided in the supplemental figures of the revised version of the manuscript (Supplemental Figure II C-H).

C3.2.2 Similarly, it would be great if the authors could perform a similar RNAseq analysis, or at least target 5 -10 of the key genes identified in the mouse study, using the primary airway endothelial cell model and plasma challenge.
R3.2.2 We thank the reviewer for his suggestion and performed gene expression analysis of Calca, Calcl, Ramp1, VEGFa and VEGFc as well as biostatistical analysis of a LPS induced sepsis human dataset to meet his concerns. The results are added to the revised manuscript in Figure 1.

C3.3.1 The recombinant procalcitonin variant study: This is an excellent component of the study and suggests that antagonizing this isoform could have potential therapeutic

benefit. That said, there are a couple of areas that need to be clarified. First, in the methods the authors refer to the transcript isoforms as variants. I am not certain this is the correct terminology as that has me thinking about SNP variants. It would probably be easier for the reader to follow if you refer to it as a splice isoform or transcript isoform consistently throughout the manuscript.

R3.3.1 In response to the reviewers remark, we exchanged the term isoform to variant isoform.

C3.3.2 More importantly, though, how does this isoform relate to human biology? I assume this isoform is also found in humans. A description of the human isoform would be helpful – how evolutionarily conserved are these isoforms in mice to humans? Evidence that the isoform can be found in human endothelial cells is necessary to demonstrate that the pathway is conserved.

R3.3.2: Procalcitonin isoforms have also been found in humans. Under inflammatory conditions, they can be found at equal parts in humans. We added the literature accordingly to the revised version of the manuscript (page 5, references number 11 and 12).

C3.3.3 Does the polyclonal antibody that was developed also antagonize the human isoform and have similar results? It would be helpful to know if the antibody developed has cross reactivity or if a new antibody needs to be developed.

R3.3.3 Human and murine procalcitonin exhibit partially different amino acid sequences and we had developed antibodies specifically targeting human truncated procalcitonin to antagonize endogenous human procalcitonin, for example in human patients plasma. For these experiments, recombinantly made truncated procalcitonin was used for evaluation of specificity (Brabenec et al., Am J Resp Crit Care Med, 2022).

C3.3.4 The authors also discuss olcegepant and inhibition of DPP4 as abolishing the IL-17 mediated response. Anything that can confirm these results in a human model would be helpful to understand the potential translational impact of inhibiting this transcriptional isoform.

R3.3.4: In human patients showing hyperprocalcitonemia postoperatively, we have detected upregulation of IL-17 expression in CD4+ cells (see below). This is also confirmed by other authors showing that DPP4 inhibition led to a shift in Th cell phenotype and subsequently lowered IL-17a expression (Pinheiro MM, Stoppa CL, Valduga CJ, et al. Sitagliptin inhibit human lymphocytes proliferation and Th1/Th17 differentiation in vitro. *Eur J Pharm Sci.* 2017;100:17-24. doi:10.1016/j.ejps.2016.12.040).

Figure 4: Left: Clusters within the CD4⁺ panel and their reactions under diabetic conditions compared to non-diabetic control patients, n= 6 vs. 7, bh- and Phenograph-algorithm. Right: IL-17-expression within the clusters

Minor concerns –

C3.4: The paragraphs are overly long. Particularly in the discussion. Breaking these up would make the manuscript easier to read.

R3.4: In response to the reviewers comment, we shortened the paragraphs of the discussion in the revised version of the manuscript.

C3.5: Line 417 – custom is spelled incorrectly.

R3.5: We thank the reviewer for noticing the mistake in spelling and corrected it accordingly.

Reviewer #4 (Remarks to the Author):

C4.0 Brabenec et al. performed bulk RNA sequencing of lung endothelial cells isolated from a mouse model of sepsis and identified transcriptomic changes that describe the impact of sepsis pathology on this critical population. This team has previously reported that a truncated form of procalcitonin disrupts vascular integrity using in vitro and in vivo studies that were supported with human clinical data from subjects that included those taking inhibitors of the procalcitonin activator DPP4. In new data reported here, the authors use a procalcitonin neutralizing antibody that was generated to test if inhibition blocks changes in the endothelial transcriptome as well as accompanying vascular defects and organ damage. This work addresses an important area of investigation. New host-based therapies are needed to improve sepsis outcomes. Beyond supportive care and antimicrobials there are few options available for clinicians to treat sepsis and septic shock. Below are some comments for clarifications to strengthen the paper prior to recommendation for publication.

The authors previously reported a role for procalcitonin in vascular pathology using a variety of experimental and observational approaches. This included mouse models of sepsis and various inhibition strategies to prove that procalcitonin is more than just a biomarker. In this study, the key new results are the transcriptomic evaluation of the lung endothelial population with respect to procalcitonin and expanded assessment of the protective role of procalcitonin inhibition on organ systems beyond the lung.

R4.0: We thank the reviewer for valuable and encouraging feedback.

C4.1. Because the transcriptomic analysis is a proposed major contribution of this study, there remain some questions that would be helpful to fully understand the data. The authors use enzymatic dissociation of the lung followed by CD45+ depletion and CD31 enrichment to generate “pure” endothelial populations. There are a number of detailed protocols (Conchinha NV 2021) that describe this procedure. Can the authors provide more detail in the methods on their protocol since it is provided “in brief”. Similarly, is there data available to show that the isolated population is pure?

R4.1: We thank the reviewer for this important remark. In response to the comment, we now describe our protocol in the methods section in full detail and included the reference in our protocol. In addition, we added our data on flow cytometry-based verification of endothelial cell purification in Supplemental Figure I A.

C4.2. Also, in the methods there is a 96h expansion of the isolated endothelial cells. Is the RNAseq performed after this 96h expansion? It is not clear as written especially since the main text on line 278 indicates that the RNAseq was done 18h after cecal puncture. This is an important detail since in vitro expansion of endothelial cells can cause phenotypic drift as measured by transcriptomics (Cleuren ACA 2019).

R4.2 We thank the author for this important remark. RNA was isolated immediately after isolation of endothelial cells from murine lungs. This is now clearly described in the revised methods section of the manuscript (page 7-8). The 96h expansion was only necessary for permeability assays.

C4.3. The generation of the antibody against the activated procalcitonin is used as a tool to neutralize the effects of procalcitonin on the endothelium. Although information is supplied in the supplement on the purified antibody using surface coating and data is shown that the truncated procalcitonin can be blocked experimentally, do the authors have data showing the specificity of the antibody? Either following overexpression and by Western blot showing the full blot or dose dependent inhibition in vitro of tyrosine phosphorylation on cadherin?

R4.3: In response to comment 3.2.1 of reviewer #3 we now show that the procalcitonin targeting the truncated procalcitonin isoform specifically blocks human sepsis plasma-induced VE-cadherin phosphorylation.

C4.4. On line 319 the authors claim that the procalcitonin antibody “reduced the expression of almost all DEGs during sepsis”. On line 419 the authors indicate that the antibody cut the number of genes in half. Figure 3D shows significant heterogeneity across the individual animals tested. Can the authors clarify the interpretation of how this is described and discussed? This is a key point since this can be interpreted that activated procalcitonin is the primary driver of lung endothelial pathology at the genomic level. This data could be strengthened by assessing impacts on endothelial transcriptomes following a complementary inhibition such as those shown in 3F for IL-17a protein.

R4.4: In response to this important reviewer remark, we rephrased the claim in the manuscript to “antagonizing procalcitonin led to a reduction of the number of differentially regulated genes in septic mice endothelium by approximately 50%” in accordance with the findings shown in Figure 3A-D (page 18 of the revised version of the manuscript).

C4.5. The authors and other groups have previously shown a causal link or correlation with vascular pathology and procalcitonin in multiple systems and models including sepsis. Therefore the gene expression analysis following procalcitonin seems to be the major new contribution here and could be a valuable dataset to understand therapeutic intervention in sepsis and translational potential. This data modality could also be a bridge to linking the CLP model of sepsis to any human pathology. Therefore, a major

deficit of the study and a missed opportunity is the lack of any transcriptomic comparison of the data generated here to publicly available human datasets. It would be valuable to include an analysis of the genes regulated by procalcitonin inhibition to those human paradigms (**endoDB**). In addition, although the authors acknowledge in the discussion that single cell RNAseq of lung has shown that it is one of the least heterogeneous organs with respect to endothelial cell diversity (Kalucka 2020), this study would be much more impactful if a single cell analysis was performed to understand the action of procalcitonin on the endothelium. If this is out of scope for this study, it would be important to discuss how some of the specific genes regulated in this study compare or align to those specific to endothelial subtypes in Kalucka 2020.

R4.5: We appreciate the reviewer's critique and agree that incorporating a comparison to a human dataset from EndoDB would significantly enhance our results. To address the reviewers remarks, we related our findings to results obtained in human endothelial cells exposed to LPS (no data was found for sepsis) and added our findings to the Manuscript (Figure 1M). For this, we searched the EndoDB database (<https://endotheliomics.shinyapps.io/endodb/>) with the term 'lung'. We obtained 29 hits, 13 were from human. Of these, 12 were from freshly isolated cells or primary cell cultures. No dataset was found for sepsis-treated lung endothelial cells. One microarray dataset reported gene expression changes after treatment of primary endothelial cells with LPS (E-GEOD-5883) at 4h, 8h and 24h post treatment and corresponding untreated controls, which we found appropriate for comparison. We downloaded the respective data from the GEO expression database (GSE5883, <https://www.ncbi.nlm.nih.gov/geo/>), and compared it our findings.

In human endothelial cells exposed to LPS, 643 probes were differentially expressed at 4h (treatment versus respective control), 827 DEPs at 8h, and 195 DEPs at 24h. We chose the 8h treatment for our comparison since it showed the strongest response. At 8h treatment, 416 unique genes (genes on the array were represented by multiple probes) were regulated, 243 up- and 173 down-regulated. 331 of these DEGs were also annotated in our datasets. Of these LPS-DEGs, 169 overlapped with DEGs from the contrast of CLP versus controls and 118 overlapped to the DEGs from the contrast CLP plus AB to controls. These results showed that our findings were very similar to the human LPS-induction study.

We then looked in detail at the top 50 up-regulated DEGs from the 8h LPS study. 36 of the LPS-DEGs were also annotated in our dataset. As shown in Figure 1M almost all LPS-DEGs genes were also up-regulated in our study, and all of these DEGs (except one) showed reduced expression in CLP plus AB treated samples. Again, these findings showed that our results were very similar to findings in LPS-treated human cells.

Minor points:

C4.6 On line 276 is there a reference that can be added to support the point that the "lung is the most susceptible organ" to sepsis induced injury?

R4.6 A reference supporting this point was added to the revised version of the manuscript (reference No. X), please also see our response to comment 2.3 of reviewer #2.

C4.7 In Figure 2E in column 4 of the Western Blot, is it correct that it is “trunc-PCT” alone, or is this with IgG as well?

R4.7: We have added the missing information to our figure, noting that the cells were also treated with control IgG.

Point-by-point-response to the reviewers comments

Dear reviewers,

we very much appreciate your effort to again revise our manuscript. Please find a point-by-point response to all of your comments below. Based on your comments, we added additional data to the manuscript to address your concerns and conducted the important corrections. Please find all changes highlighted in the tracked-changes version of the manuscript.

Reviewer #1 (Remarks to the Author):

The authors have revised the manuscript and figures, and included additional experimental data, in an effort to enhance the overall quality, clarity and translational value of the article under review. These efforts are recognised and appreciated. Nevertheless, I believe that my primary and most substantial concerns regarding the manuscript have largely gone unaddressed. The role of procalcitonin in the pathophysiology of sepsis has been extensively characterised in the literature, particularly with regard to the inhibition of Calcrl-signalling by the antagonist BIBN4096, which has been shown to improve survival in mouse models. Furthermore, it has been convincingly demonstrated that the observed survival benefit of Calcrl-signalling antagonism is mechanistically mediated via the induction of IL-17A, as reported by Baranowsky et al. (2021). It is important to note that while this proposed therapeutic intervention was found to be promising in small animal models, it failed translation in a more clinically relevant large animal model of porcine polymicrobial sepsis, as documented by Messerer et al. (2022). Consequently, the novel, clinically applicable information that is provided by the current manuscript remains unclear, especially due to the absence of a clearly defined clinical outcome measure. If the authors wanted to claim that their study provides meaningful novel insight of clinical relevance, they would have to provide compelling comparative data demonstrating that the truncated version of their novel procalcitonin antibody is significantly superior to the known CALCR antagonist BIBN4096. In the absence of such evidence, it is challenging to substantiate the claim of novelty or clinical translatability. In summary, I believe that the present manuscript makes only a negligible contribution to the current understanding of the role of procalcitonin in sepsis and falls short of providing substantial translational or clinical value and should therefore be published in a more specialized journal.

Response: We thank the reviewer for raising the important point regarding the effect of the procalcitonin antibody on clinical outcome measures and the probability of survival. Based on your comments, we analyzed all scoring of health status, behavior and sepsis criteria that we were demanded to conduct on an hourly basis throughout all mouse experiments based on the animal protocol approved by the authorities. We provide this data as Figure 6 in the revised version of the manuscript and a detailed description of the scoring system in the Methods. This data shows that the health status, behavior and sepsis criteria are significantly improved already during early courses of the disease and continue to diverge showing beneficial effects of the anti-procalcitonin antibody (Figure 6a-c). The overall score at 18h post sepsis-induction equally verifies the beneficial effect of the antibody (Figure 6d).

As a score of 20 in any category was defined as the humane endpoint for mice in these experiments, we conducted a Kaplan-Meier-analysis plotting the probability of survival. Please find the results included in the revised version of the manuscript as Figure 6e. This data shows a significant difference between septic animals treated with control-IgG and septic mice treated with the procalcitonin-targeting antibody.

To address the reviewers concerns regarding the comparability with the CRLR/RAMP1-inhibitor BIBN4096, we plotted the data regarding the reach of the humane endpoint of a score of 20 in any category during the course of sepsis from a previous study on the use of BIBN4096 in septic mice. Interestingly, we can clearly show that the beneficial effect of the antibody specifically targeting truncated procalcitonin shows superior effects on murine survival during sepsis compared to BIBN4096 that did not show a significant benefit vs. vehicle-control (Fig. 6f).

BIBN4096 is a substance clinically licensed as an anti-migraine treatment known to exert possible side effects (for example, potentially liver toxicity). CGRP exerts various actions on the vasculature and has recently been shown to ameliorate sepsis-induced intestinal injury (Ning W et al., 2023, <https://doi.org/10.1016/j.intimp.2023.109747>). These aspects could render the inhibition of this pathway potentially not beneficial during sepsis. In contrast, the procalcitonin antibody appears to not exert adverse effects but ameliorate sepsis severity and beneficially influence murine outcomes. This underlines the novelty of the presented strategy using procalcitonin-targeting antibodies to inhibit the procalcitonin signaling pathway.

Reviewer #2 (Remarks to the Author):

I appreciate that the authors have made a diligent revision of the manuscript. In the revised version, the authors have addressed all my comments appropriately and further enhanced the manuscript. The manuscript has been improved at a reasonably significant level.

Reviewer #3 (Remarks to the Author):

The authors have addressed all my comments.

Reviewer #4 (Remarks to the Author):

R4.1: Thank you for the added data.

R4.2: Thank you for the clarification

R4.3: Thank you for the added data

R4.4: Thank you for the clarification

R4.5: I thank the authors for the addition of the human transcriptomics analysis.

R4.6: Thank you for the added references.

R4.7: Thank you for the correction.

Minor Comments:

I have noted a number of spelling corrections needed in the text and figure legends. Please do a final check.

Response: Thank you very much for this remark, we corrected multiple spelling errors throughout the manuscript.

The addition of the human cohort and human in vitro studies strengthens the manuscript. Please ensure that references to plasma and serum are used correctly throughout the report. The legends and methods indicate that serum was collected from the human subjects, but it seems that in the discussion of the results you indicate that plasma was used. Please double check. Also, can you indicate in the methods more specifically how experiments were

performed with mouse or human serum (plasma) with respect to percent of human or mouse serum added to the cell culture media? Was FBS still included in these experiments?

Response: Thank you very much for your comment. We revised the use of the words serum and plasma carefully throughout the manuscript – for procalcitonin concentrations in mice, plasma was used. For human samples, serum was used. We also specified the Methods regarding the in vitro experiments in the revised version of the manuscript. For these experiments, we used plasma from septic or control mice diluted 1:1 in serum-free cell culture media (see Methods in the revised version of the manuscript).

Point-by-point response to reviewers comments

Dear reviewer,

we very much appreciate your effort to again revise our manuscript. Please find a point-by-point response to all of your comments below. Based on your comments, we revised our manuscript to address your concerns and conducted the important corrections. Please find all changes highlighted in the tracked-changes version of the manuscript.

Reviewer #2 (Remarks to the Author):

Comment 1: The manuscript has been strengthened by the Kaplan-Meier survival curves showing that neutralizing procalcitonin, but not BIBN4096, was able to improve mice after CLP surgery. Please mention that log rank test or Gehan-Breslow-Wilcoxon test was used for comparing Kaplan-Meier survival curves in the Statistical analysis section or in the legend of Fig. 6.

Response 1: We sincerely thank the reviewer for drawing our attention to the omission of statistical information. We have analysed the survival data using the log-rank test and have included this information in the revised manuscript, both in the Statistical Analysis section and in the legend of Figure 6.

Comment 2: In addition, there are still open questions regarding the CLP model's multifactorial determinants of its reproducibility. The CLP model can be modified to produce varying degrees of severity, depending on the cecum ligation site and the number of punctures made. Thus, variation in the number and size of the puncture(s) can be used to modulate sepsis severity, inflammatory response, and mortality rate. Accordingly, the authors should clearly mention how many centimeters from the cecum tip they tightly ligated.

Response 2: We thank the reviewer for this insightful comment and fully agree that variability can occur in the CLP model due to its multifactorial determinants. As the reviewer correctly points out, we also took particular care to minimize such variability in our experiments. Specifically, we ligated the cecum 14 mm from the cecal tip in each mouse, and to further reduce technical variation, all CLP surgeries were performed by one experienced researcher blinded to the experimental group to avoid technical variations. We added this information to our manuscript.

In addition, we added supplemental figure II, showing that bacterial colony forming units from peritoneal lavages show no significant differences in bacterial load in CLP groups. We think this information is crucial showing that the insult of sepsis induction did not vary in mean between groups (see below).

Supplementary Figure II.

Suppl. Fig. II. A Representative photographs showing agar plates of murine blood after incubation at 37°C for 24hours. **B** Colony forming unit count in peritoneal lavange, culture withdrawn 18hours after sepsis induction by cecal ligation and puncture (CLP)/control (Sham) following injection of the antibody or respective control IgG, n=5-11 mice/group.

Comment 3: Also, the authors may describe that their mice after CLP surgery were all fatal (although, in this study, CLP mice appears to be nearly fatal, and it is a bit concerning that there is an apparent difference between the CLP+IgG group (Fig. 6e) and the CLP+DMSO group (Fig. 6f)). For instance, in our laboratory, all mice subjected to CLP without treatment die within 48 hours. Moreover, we have only one surgeon for CLP surgery in one study to avoid technical variations.

Response 3: Regarding the reviewer's concern about the apparent difference between the CLP+IgG and CLP+DMSO groups, we have now plotted the survival curves together and found no statistically significant differences between these groups. As noted, CLP-treated animals are typically fatal within 48 hours; however, in our study, the observation period was limited to the first 18 hours post-surgery. As mentioned above, all CLP surgeries were performed by a single experienced researcher to minimize variability.

Comment 4: Finally, the authors may describe behavioral characteristics of CLP mice. Thus, mice after CLP surgery were all lethargic, showed lack of interest in their environment, displayed piloerection, and had crusty exudates around their eyes. These points would provide more evidence that great care was taken to minimize variability in disease severity.

Response 4: In terms of behavioral characteristics, following CLP surgery, the mice displayed lethargy and reduced mobility, showed no interest in their surroundings, and did not engage in social behaviors such as sniffing or grooming. These observations are consistent with the typical clinical signs of sepsis. We added these observations to our manuscript.